# Unified Covariate Adjustment for Causal Inference

**Yonghan Jung**[1], **Jin Tian**[2], and **Elias Bareinboim**[3]

[1]Purdue University
[2]Mohamed bin Zayed University of Artificial Intelligence
[3]Columbia University
[1]jung222@purdue.edu, [2]jin.tian@mbzuai.ac.ae, [3]eb@cs.columbia.edu

## Abstract

Causal effect identification and estimation are fundamental tasks found throughout the data sciences. Although causal effect identification has been solved in theory, many existing estimators only address a subset of scenarios, known as the sequential back-door adjustment (SBD) (Pearl and Robins, 1995a) or g-formula (Robins, 1986). Recent efforts for developing general-purpose estimators with broader coverage, incorporating the front-door adjustment (FD) (Pearl, 2000) and others, are not scalable due to the high computational cost of summing over a high-dimensional set of variables. In this paper, we introduce a novel approach that achieves broad coverage of causal estimands beyond the SBD, incorporating various sum-product functionals like the FD, while achieving scalability – estimated in polynomial time relative to the number of variables and samples in the problem. Specifically, we present the class of *unified covariate adjustment (UCA)* for which we develop a scalable and doubly robust estimator. In particular, we illustrate the expressiveness of UCA for a wide spectrum of causal estimands (e.g., SBD, FD, and others) in causal inference. We then develop an estimator that exhibits computational efficiency and double robustness. Experiments corroborate the scalability and robustness of the proposed framework.

## 1 Introduction

Causal inference is a crucial aspect of scientific research, with broad applications ranging from social sciences to economics, and from biology to medicine. Two significant tasks in causal inference are causal effect identification and estimation. *Causal effect identification* concerns determining the conditions under which the causal effect can be inferred from a combination of available data distributions and a causal graph depicting the data-generating process. *Causal effect estimation*, on the other hand, develops an estimator for the identified causal effect expression using finite samples.

Causal effect identification theories have been well-established across various scenarios. These include cases where the input distribution is purely observational (Tian and Pearl, 2003; Shpitser and Pearl, 2006; Huang and Valtorta, 2006) (known as *observational identification* or obsID) or a combination of observational and interventional (Bareinboim and Pearl, 2012a; Lee et al., 2019) (referred to as *generalized identification* or gID); scenarios where the target query and input distributions originate from different populations (Bareinboim and Pearl, 2012b; Bareinboim et al., 2014; Bareinboim and Pearl, 2016; Correa et al., 2018; Lee et al., 2020) (known as *recoverability* or *transportability*); or cases where the target query is *counterfactual* (Rung 3) (Correa et al., 2021) (referred to as Ctf-ID) beyond interventional (Rung 2) of the *Ladder of Causation* (Pearl and Mackenzie, 2018; Bareinboim et al., 2020). In these situations, algorithmic solutions have been devised that take input distributions along with specified target queries and formulate identification functionals as arithmetic operations (sums/integration, products, ratios) on conditional distributions induced from input distributions.

38th Conference on Neural Information Processing Systems (NeurIPS 2024).

Despite all the progress, existing estimators cover only a subset of all identification scenarios. Specifically, well-established estimators for the back-door (BD) adjustment (Pearl, 1995), represented as $\sum_z \mathbb{E}[Y \mid x, z]P(z)$, and sequential back-door adjustment (SBD) (Robins, 1986; Pearl and Robins, 1995b) and off-policy evaluations (OPE) (Murphy, 2003), which is an SBD with policy interventions, are known for their robustness to the bias (Bang and Robins, 2005; Robins et al., 2009; van der Laan and Gruber, 2012; Murphy, 2003; Rotnitzky et al., 2017; Luedtke et al., 2017; Uehara et al., 2022; Díaz et al., 2023). These estimators are also *scalable*; i.e., evaluable in polynomial time relative to the number of covariates ($|Z|$) and capable in the presence of mixed discrete and continuous covariates. However, SBDs only address a fraction of the broader spectrum of identification scenarios.

Beyond SBD, recent efforts have expanded to developing estimators for the front-door (FD) adjustment $\sum_{z,x'} \mathbb{E}[Y \mid x', z]P(z \mid x)P(x')$ (Pearl, 1995). At first glance, this adjustment appears similar to SBD, as both involve the sum-product of conditional probabilities. However, FD involves treatments variables in dual roles – one being summed ($x'$ in $\sum_{x'} \mathbb{E}[Y \mid x', z]P(x')$) and the other being fixed ($x$ in $P(z \mid x)$). While FD estimators achieving doubly robustness have been developed (Fulcher et al., 2019; Guo et al., 2023), they lack scalability due to the necessity of summing over the values of $Z$ (i.e., $\sum_z$), thereby limiting its practicality when $Z$ is high-dimensional or continuous.

Similar challenges arise in more general identification scenarios beyond SBD and FD. Recent efforts have focused on developing estimators for broad causal estimands, such as *Tian's adjustment* (Tian and Pearl, 2002a), which incorporates FD and other cases where causal effects are represented as sum-product functionals (Bhattacharya et al., 2022). These efforts also include work on covering any identification functional (Jung et al., 2021a; Xia et al., 2021, 2022; Bhattacharya et al., 2022; Jung et al., 2023a). While these estimators are designed to achieve a wide coverage of functionals, they lack scalability due to the necessity of summing over high-dimensional variables.

Thus far, we have assessed the pair (functional class, estimator) based on two criteria: (1) *coverage* of the functional class, and (2) *scalability* of the corresponding estimators. Scalable estimators achieving doubly robustness have been established predominantly for BD/SBD classes. While recent studies have developed estimators with a strong emphasis on coverage (e.g., any identification functional), less attention has been given to achieving scalability.

In this paper, we establish a novel pair of a functional class and its corresponding estimation frameworks designed to ensure scalability while covering a broad spectrum of identification functionals. Our work aims to maximize coverage, enabling the effective development of scalable estimators with the doubly robust prop-

| Function class | Coverage | | Scalability | |
|---|---|---|---|---|
| | **Prior** | **UCA** | **Prior** | **UCA** |
| **BD/SBD/OPE** | ✓ | ✓ | ✓ | ✓ |
| **FD** | ✓ | ✓ | ✗ | ✓ |
| **Tian's** | ✓ | ✓ | ✗ | ✓ |
| **obsID/gID** | ✓ | ▲ | ✗ | ✓ |
| **Ctf-ID** | ▲ | ▲ | ? | ✓ |
| **Transportability** | ▲ | ▲ | ? | ✓ |

Table 1: Scope. ✓ denotes the addressed area (by UCA or prior works). ✗ denotes the unaddressed area. ▲ denotes the partially addressed area. ? indicates areas where no known results are present.

erty. This functional class, termed *unified covariate adjustment* (UCA), integrates a sum-product of conditional distributions appearing in many causal inference scenarios such as BD/FD, Tian's adjustment, $S$-admissibility in transportability/recoverability (Bareinboim and Pearl, 2016), effect-of-treatment-on-the-treated (ETT) (Heckman, 1992), and nested counterfactuals (Correa et al., 2021). The coverage of the proposed class is further demonstrated through the application to a novel estimand for the counterfactual directed effect (Ctf-DE) derived from fairness analysis (Plečko and Bareinboim, 2024). For the proposed UCA class, we develop a scalable and doubly robust estimator evaluable computationally efficiently relative to the number of samples. Table 1 visualizes the scope of our framework. The contributions of this paper are as follows:

1. We introduce *unified covariate adjustment (UCA)*, a comprehensive framework that encompasses a broad class of sum-product causal estimands. This framework's expressiveness is demonstrated across various scenarios beyond SBD, including Tian's adjustment that incorporates FD and others as well novel counterfactual scenarios in fairness analysis.

2. We develop a corresponding estimator that is computationally efficient and doubly robust and provide its finite sample guarantee. We demonstrate scalability and robustness to bias both theoretically and empirically through simulations.

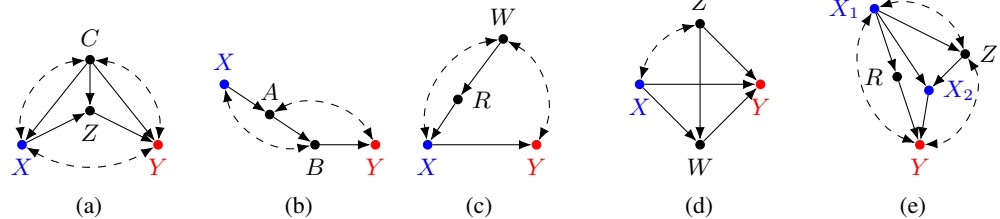

Figure 1: **(a)** Front-door in Example 1, **(b)** Verma in Example 2, **(c)** Napkin, **(d)** Standard fairness model in Example 3, and **(e)** Example graph from (Jung et al., 2021a, Fig. 1b)

**Notations.** We use $(\mathbf{X}, X, \mathbf{x}, x)$ to denote a random vector, variable, and their realized values, respectively. For a function $f(\mathbf{z}_i)$ for $i = 1, 2, \cdots,$, we use $\sum_i f(\mathbf{z}_i) = f(\mathbf{z}_1) + f(\mathbf{z}_2) \cdots$. Also, for a function $f(\mathbf{z})$, we use $\sum_{\mathbf{z}} f(\mathbf{z})$ to denote the summation/integration over a mixture of discrete/continuous random variables $\mathbf{Z}$. For example, we write the back-door adjustment as $\sum_{\mathbf{z}} \mathbb{E}_P[Y \mid x, \mathbf{z}]P(\mathbf{z})$ when $\mathbf{Z}$ is a mixture of discrete/continuous variables. Given an ordered set $\mathbf{X} = \{X_1, \cdots, X_n\}$, we denote $\mathbf{X}^{(i)} := \{X_1, \cdots, X_i\}$ and $\mathbf{X}^{\geq i} := \{X_{i+1}, \cdots, X_m\}$ for $m = |\mathbf{X}|$. For a discrete $\mathbf{X}$, we use $\mathbb{1}_{\mathbf{x}}(\mathbf{X})$ as a function such that $\mathbb{1}_{\mathbf{x}}(\mathbf{X}) = 1$ if $\mathbf{X} = \mathbf{x}$; $\mathbb{1}_{\mathbf{x}}(\mathbf{X}) = 0$ otherwise. $P(\mathbf{V})$ denotes a distribution over $\mathbf{V}$ and $P(\mathbf{v})$ as a probability at $\mathbf{V} = \mathbf{v}$, We use $\mathbb{E}_P[f(\mathbf{V})]$ and $\mathbb{V}_P[f(\mathbf{V})]$ to denote the mean and variance of $f(\mathbf{V})$ relative to $P(\mathbf{V})$. We use $\|f\|_P := \sqrt{\mathbb{E}_P[\{f(\mathbf{V})\}^2]}$ as L2-norm of $f$ with $P$. If a function $\widehat{f}$ is a consistent estimator of $f$ having a rate $r_n$, we will use $\widehat{f} - f = o_P(r_n)$. We will say $\widehat{f}$ is $L_2$-consistent if $\|\widehat{f} - f\|_P = o_P(1)$. We will use $\widehat{f} - f = O_P(1)$ if $\widehat{f} - f$ is bounded in probability. Also, $\widehat{f} - f$ is said to be bounded in probability at rate $r_n$ if $\widehat{f} - f = O_P(r_n)$. $[n] := \{1, \cdots, n\}$ is a collection of index. $\mathcal{D} := \{\mathbf{V}_{(i)} : i \in [n]\}$ denotes a sample set, where $\mathbf{V}_{(i)}$ denote the $i$th sample in $\mathcal{D}$. The empirical average of $f(\mathbf{V})$ with samples $\mathcal{D}$ is $\mathbb{E}_{\mathcal{D}}[f(\mathbf{V})] := (1/|\mathcal{D}|) \sum_{i:\mathbf{V}_{(i)} \in \mathcal{D}} f(\mathbf{V}_{(i)})$.

**Structural causal models.** We use structural causal models (SCMs) (Pearl, 2000; Bareinboim et al., 2020) as our framework. An SCM $\mathcal{M}$ is a quadruple $\mathcal{M} = \langle \mathbf{U}, \mathbf{V}, P(\mathbf{U}), \mathcal{F} \rangle$, where $\mathbf{U}$ is a set of exogenous (latent) variables following a joint distribution $P(\mathbf{U})$, and $\mathbf{V}$ is a set of endogenous (observable) variables whose values are determined by functions $\mathcal{F} = \{f_{V_i}\}_{V_i \in \mathbf{V}}$ such that $V_i \leftarrow f_{V_i}(\mathbf{pa}_i, \mathbf{u}_i)$ where $\mathbf{PA}_i \subseteq V$ and $\mathbf{U}_i \subseteq \mathbf{U}$. Each SCM $\mathcal{M}$ induces a distribution $P(\mathbf{V})$ and a causal graph $\mathcal{G} = \mathcal{G}(\mathcal{M})$ over $\mathbf{V}$ in which directed edges exists from every variable in $\mathbf{PA}_i$ to $V_i$ and dashed-bidirected arrows encode common latent variables. Performing an intervention fixing $\mathbf{X} = \mathbf{x}$ is represented through the do-operator, $\mathrm{do}(\mathbf{X} = \mathbf{x})$, which encodes the operation of replacing the original equations of $X$ (i.e., $f_X(\mathbf{pa}_x, \mathbf{u}_x)$) by the constant $x$ for all $X \in \mathbf{X}$ and induces an interventional distribution $P(\mathbf{V} \mid \mathrm{do}(\mathbf{x}))$. For any $\mathbf{Y} \subseteq \mathbf{V}$, the *potential response* $\mathbf{Y}_{\mathbf{x}}(\mathbf{u})$ is defined as the solution of $\mathbf{Y}$ in the submodel $\mathcal{M}_{\mathbf{x}}$ given $\mathbf{U} = \mathbf{u}$, which induces a *counterfactual variable* $\mathbf{Y}_{\mathbf{x}}$.

**Related work.** Our work is an extension of existing sequential back-door adjustment (SBD) estimators (Mises, 1947; Bickel et al., 1993; Bang and Robins, 2005; Robins et al., 2009; van der Laan and Gruber, 2012; Rotnitzky et al., 2017; Luedtke et al., 2017; Díaz et al., 2023) to a broader class of sum-product functionals, such as the front-door adjustment (FD) and Tian's adjustment (Tian and Pearl, 2002a) which generalizes FD and more, and nested counterfactuals, which will be detailed in later sections. Our work is aligned with recent works of Chernozhukov et al. (2022); Li and Luedtke (2023); Quintas-Martinez et al. (2024), which examined SBD derived from various joint distributions. Specifically, Li and Luedtke (2023) considered the SBD setting where conditional distributions are induced from different sources. In contrast, we study a broader class of sum-product functionals from multiple populations. Also, Quintas-Martinez et al. (2024) considered the Markovian model $\prod_{i=1}^{n} P^i(V_i \mid \mathbf{PA}_i)$ where each $P^i$ can be distinct. In contrast, we study a broader class of estimands that are not confined to conditioning on $\mathbf{PA}_i$. On the other hands, (Chernozhukov et al., 2022) considered the case where covariate distributions are allowed to be changed, and demonstrated that FD can be captured through this technique. Our work expands on these findings by covering a broader class, such as the Tian's adjustment and a nested counterfactual in fairness literature, and by providing a more formal theory that includes finite sample guarantees and asymptotic analysis.

## 2 Unified Covariate Adjustment

A class of causal estimands termed *unified covariate adjustment (UCA)* is defined as follows:

**Definition 1** (**Unified Covariate Adjustment (UCA)**). *Let $\Psi[\mathbf{P}; \boldsymbol{\sigma}]$ denote the following probability measure over an ordered set $\mathbf{V} := (\mathbf{C}_1, \mathbf{R}_1, \cdots, \mathbf{C}_m, \mathbf{R}_m, Y := \mathbf{C}_{m+1})$: $\Psi[\mathbf{P}; \boldsymbol{\sigma}] := P^{m+1}(Y \mid \mathbf{S}_m) \prod_{i=1}^{m} P^i(\mathbf{C}_i \mid \mathbf{S}_{i-1}) \sigma_{\mathbf{R}_i}^i(\mathbf{R}_i \mid \mathbf{S}_i \setminus \mathbf{R}_i)$, where*

- $\mathbf{P} := \{P^i(\mathbf{V}) : i \in [m+1]\}$ *is a set of distributions in the form of $P^i(\mathbf{V}) = Q^i(\mathbf{V} \mid \mathbf{S}_{i-1}^b = \mathbf{s}_{i-1}^b)$, where $Q^i$ is a distribution, $\mathbf{S}_{i-1}^b$ is a (potentially empty) set. Each pairs $P^i(\mathbf{V})$ and $P^j(\mathbf{V})$ can be the same ($P^i(\mathbf{V}) = P^j(\mathbf{V})$) or distinct ($P^i(\mathbf{V}) \neq P^j(\mathbf{V})$).*

- *For $i \in [m+1]$, $\mathbf{S}_{i-1} := (\mathbf{C}^{(i-1)} \cup \mathbf{R}^{(i-1)}) \setminus \mathbf{S}_{i-1}^b$.*

- *Each $\mathbf{R}_i$ is controlled by a pre-specified / known probability measure $\sigma_{\mathbf{R}_i}^i := \sigma_{\mathbf{R}_i}^i(\mathbf{r}_i \mid \mathbf{s}_i \setminus \mathbf{r}_i)$ where $\sum_{\mathbf{r}_i} \sigma_{\mathbf{R}_i}^i(\mathbf{r}_i \mid \mathbf{s}_i \setminus \mathbf{r}_i) = 1$ and $0 \leq \sigma_{\mathbf{R}_i}^i \leq 1$ almost surely (e.g., $\sigma_{\mathbf{R}_i}^i := \mathbb{1}_{\mathbf{r}_i}(\mathbf{R}_i)$).*

*Then, the expectation of $Y$ over $\Psi[\mathbf{P}; \boldsymbol{\sigma}]$ is called a **Unified Covariate Adjustment (UCA)**:*

$$\psi_0 := \mathbb{E}_{\Psi[\mathbf{P}; \boldsymbol{\sigma}]}[Y] = \sum_{\mathbf{c} \cup \mathbf{r}} \mathbb{E}_{P^{m+1}}[Y \mid \mathbf{s}_m] \prod_{i=1}^{m} P^i(\mathbf{c}_i \mid \mathbf{s}_{i-1}) \sigma_{\mathbf{R}_i}^i(\mathbf{r}_i \mid \mathbf{s}_i \setminus \mathbf{r}_i). \tag{1}$$

We will exemplify that UCA encompasses many well-known causal estimands, including the sequential back-door adjustment (SBD) (Robins, 1986; Pearl and Robins, 1995b), front-door adjustment (Pearl, 1995), Tian's adjustment (Tian and Pearl, 2002a), $S$-admissibility in transportability/recoverability (Bareinboim and Pearl, 2016), effect-of-treatment-on-the-treated (ETT) (Heckman, 1992), nested counterfactuals (Correa et al., 2021), treatment-treatment interaction (Jung et al., 2023b), and off-policy evaluation (Murphy, 2003). This section particularly focuses on recently developed and lesser-known estimands, for which scalable estimators have been rarely explored. Appendix B provides additional examples, demonstrating how UCA can represent well-known estimands such as off-policy evaluation and $S$-admissibility.

At first glance, UCA closely resembles the sequential back-door adjustment (SBD) (Robins, 1986; Pearl and Robins, 1995b). Indeed, UCA is reduced to SBD in the special case where $P^i = P(\mathbf{V})$ for all $i = 1, \cdots, m+1$ and $\sigma_{\mathbf{R}_i}^i := \mathbb{1}_{\mathbf{r}_i}(\mathbf{R}_i)$; i.e., $\psi_0 = \sum_{\mathbf{c}} \mathbb{E}_P[Y \mid \mathbf{c}^{(m)} \cup \mathbf{r}^{(m)}] \prod_{i=1}^{m} P(\mathbf{c}_i \mid \mathbf{c}^{(i-1)} \cup \mathbf{r}^{(i-1)})$. However, UCA provides flexibility to represent target estimands beyond SBD by allowing $P^i$ to be any distribution that aligns with the target estimand, permitting arbitrary conditional distributions beyond the observational distribution $P$. To demonstrate, consider the front-door adjustment (FD) scenario (Pearl, 1995) depicted in Fig. 1a:

$$\mathbb{E}[Y \mid \text{do}(x)] = \sum_{c,z,x'} \mathbb{E}[Y \mid c, x', z] P(z \mid c, x) P(c, x'). \tag{2}$$

Even though FD cannot be expressed using SBD because the treatment variable $X$ is being fixed (in $P(z \mid c, x)$) and summed (with $\sum_{x'}$) simultaneously, it can be represented through UCA as follows:

**Example 1** (**FD as UCA**). *FD can be written as the expectation of $Y$ over $P(Y \mid Z, X, C) P(Z \mid x, C) P(X, C)$. We set $\mathbf{C}_1 := \{X, C\}$, $\mathbf{C}_2 := \{Z\}$, $\mathbf{R} = \emptyset$, $P^1(\mathbf{C}_1) = P(X, C)$, $P^2(\mathbf{C}_2 \mid \mathbf{S}_1) = P(Z \mid x, C)$ with $\mathbf{S}_1^b = \{X\}$, $\mathbf{S}_1 = \{C\}$, and $P^3(Y \mid \mathbf{S}_3) = P(Y \mid Z, X, C)$ with $\mathbf{S}_2 = \{Z, X, C\}$.*

Next, consider *Verma's equation* (Verma and Pearl, 1990; Tian and Pearl, 2002b) with Fig. 1b:

$$\mathbb{E}[Y \mid \text{do}(x)] = \sum_{b,a,x'} \mathbb{E}[Y \mid b, a, x] P(b \mid a, x') P(a \mid x) P(x'), \tag{3}$$

where $X$ is fixed to $x$ in $\mathbb{E}[Y \mid x, a, b]$ and $P(a \mid x)$ while summed in $P(b \mid a, x')$ and $P(x')$. Similar to FD, due to the dual role of $X$, the existing SBD framework is not suitable to express Verma's equation, which can be represented through UCA as follows:

**Example 2** (**Verma as UCA**). *Verma's equation is expressible as the expectation of $Y$ over $P(Y \mid B, A, x) P(B \mid A, X) P(A \mid x) P(X)$. We set $\mathbf{C}_1 = \{X\}$, $\mathbf{C}_2 = \{A\}$, $\mathbf{C}_3 = \{B\}$, and $\mathbf{R} = \emptyset$. We map $P^1(\mathbf{C}_1) := P(X)$, $P^2(\mathbf{C}_2 \mid \mathbf{S}_1) = P(A \mid x)$ with $\mathbf{S}_1 = \emptyset$, $\mathbf{S}_1^b = \{X\}$, $P^3(\mathbf{C}_3 \mid \mathbf{S}_2) = P(B \mid A, X)$ with $\mathbf{S}_2 = \{A, X\}$, and $P^4(Y \mid \mathbf{S}_3) = P(Y \mid B, A, x)$ with $\mathbf{S}_3 = \{B, A\}$, $\mathbf{S}_3^b = \{X\}$.*

In both examples, the variable $\mathbf{S}^b i = X$ is *bifurcated*, being fixed in some conditional distributions (e.g., $P(z \mid x, c)$ in the front-door criterion (FD)) and summed over $\sum x'$ in others (e.g., $P(y \mid z, x', c)$

**Algorithm 1:** `Tian-to-UCA`$(\mathcal{G}, \mathbf{V} := (V_1, \cdots, V_K, Y))$

---

**Input:** A graph $\mathcal{G}$ and a set of topologically ordered variables $\mathbf{V} := (V_1, \cdots, V_K, Y)$.

1 Set $\mathbf{C}_1 := (V_1, \cdots, V_{k-1}, V_k := X, V_{k+1}, \cdots, V_{k+i_1})$ as an ordered sequence, where $(V_1, \cdots, V_{k-1})$ are predecessors of $X$, and $(V_{k+1}, \cdots, V_{k+i_1})$ are successors of $X$ within $\mathbf{S}_X$.

2 Set $P^1(\mathbf{C}_1) := P(\mathbf{C}_1)$, $i := 2$ and $\mathbf{R} := \emptyset$.

3 **while** $\mathbf{V} \setminus (\{Y\} \cup \mathbf{C}^{(i-1)}) \neq \emptyset$ **do**

4      **if** $\mathbf{C}_{i-1} \subseteq \mathbf{S}_X$, set $\mathbf{C}_i$ as the next sequence of vertices in $\mathbf{V} \setminus (\{Y\} \cup \mathbf{C}^{(i-1)})$ that are not in $\mathbf{S}_X$; $\mathbf{S}_{i-1} := \mathbf{C}^{(i-1)} \setminus \{X\}$; and $P^i(\mathbf{C}_i \mid \mathbf{S}_{i-1}) := P(\mathbf{C}_i \mid \mathbf{S}_{i-1}, x)$ with $\mathbf{S}_{i-1}^b := \{X\}$.

5      **else**, set $\mathbf{C}_i$ as the next sequence of vertices in $\mathbf{V} \setminus (\{Y\} \cup \mathbf{C}^{(i-1)})$ that are in $\mathbf{S}_X$; $\mathbf{S}_{i-1} := \mathbf{C}^{(i-1)}$; and $P^i(\mathbf{C}_i \mid \mathbf{S}_{i-1}) := P(\mathbf{C}_i \mid \mathbf{S}_{i-1})$ with $\mathbf{S}_{i-1}^b := \emptyset$.

6      $i \leftarrow i + 1$.

7 **end**

8 Set $m \leftarrow i$. If $Y \in \mathbf{S}_X$, set $\mathbf{S}_m := \mathbf{C}^{(m)}$, $\mathbf{S}_m^b = \emptyset$, and $P^{m+1}(Y \mid \mathbf{S}_m) = P(Y \mid \mathbf{S}_m)$. Otherwise, set $\mathbf{S}_m := \mathbf{C}^{(m)} \setminus \{X\}$, $\mathbf{S}_m^b = \{X\}$, and $P^{m+1}(Y \mid \mathbf{S}_m) = P(Y \mid \mathbf{S}_m, x)$.

9 **return** $\mathbb{E}_{\Psi[\mathbf{P}]}[Y]$ *where* $\Psi[\mathbf{P}] := P^{m+1}(Y \mid \mathbf{S}_m) \prod_{i=1}^m P^i(\mathbf{C}_i \mid \mathbf{S}_{i-1})$.

---

in FD). Both FD and Verma's equations are special cases of *Tian's adjustment* (Tian and Pearl, 2002a), which states that $\mathbb{E}[Y \mid \mathrm{do}(x)]$ is identifiable under certain conditions. Specifically, when $X$ and its children $\mathrm{ch}\,\mathcal{G}(X)$ in the graph $\mathcal{G}$ are not connected by bidirected edges, it can be expressed as:

$$\mathbb{E}[Y \mid \mathrm{do}(x)] = \sum_{\mathbf{v} \setminus xy} \sum_{x'} \mathbb{E}_{P'}[Y \mid \mathbf{v}^{(K)}] \prod_{i=1}^K P'(v_i \mid \mathbf{v}^{(i-1)}), \tag{4}$$

where $\mathbf{V} := (V_1, V_2, \cdots, V_K, Y)$ is a topologically ordered set with $V_k := X$ for some $k$ being the treatment variable $X$, $P'(v_i \mid \mathbf{v}^{(i-1)}) := P(v_i \mid \mathbf{v}^{(k-1)}, x, v_{k+1}, \cdots, v_{i-1})$ (i.e., $X$ is fixed to $x$) if $V_i \notin \mathbf{S}_X$ where $\mathbf{S}_X$ is the set of vertices connected with $X$ through bidirected edges, and $P'(v_i \mid \mathbf{v}^{(i-1)}) := P(v_i \mid \mathbf{v}^{(k-1)}, x', v_{k+1}, \cdots, v_{i-1})$ (i.e., $X$ is summed with $\sum_{x'}$) if $V_i \in \mathbf{S}_X$. In Tian's adjustment, $X$ is bifurcated into *summed* through $\sum_{x'}$ and *fixed* to $X = x$. We exhibit the expressiveness of UCA for Tian's adjustment:

**Proposition 1.** *Tian's adjustment in Eq.* (4) *is UCA-expressible through Algo. 1.*

Next, we exhibit the coverage of the UCA for a counterfactual quantity in the fairness literature. Specifically, we focus on the counterfactual directed effect (Ctf-DE) in the *Standard fairness model* (SFM) (Plečko and Bareinboim, 2024), as illustrated in Fig. 1d. This model includes several key components: the protected (discrete) attribute ($X$), such as race; the baseline covariates ($Z$), like age; the mediator variables ($W$) affected by $X$, for example, educational level; and the outcome variable ($Y$), such as salary. Consider a scenario where we investigate the the query, "*What would be the expected salary for someone who is Black, but hypothetically of Asian race and had been educated as a White person typically would be?*". The query is represented as Ctf-DE: $\mathbb{E}[Y_{X=x_0, W_{X=x_1}} \mid X = x_2]$, where $x_0$, $x_1$, and $x_2$ correspond to the races Asian, White, and Black, respectively. This query can be identified through the algorithm in (Correa et al., 2021) under the SFM in Fig. 1d:

$$\mathbb{E}[Y_{X=x_0, W_{X=x_1}} \mid X = x_2] = \sum_{w,z} \mathbb{E}[Y \mid X = x_0, w, z] P(w \mid X = x_1, z) P(z \mid X = x_2). \tag{5}$$

This identification functional is UCA-expressible:

**Example 3 (Ctf-DE as UCA).** *The Ctf-DE is expressible through the expectation of $Y$ over $P(Y \mid X = x_0, W, Z) P(W \mid X, Z) P(Z \mid X = x_2) \mathbb{1}_{x_1}(X)$. Set $\mathbf{R}_1 := \{X\}$, $\sigma_{\mathbf{R}_1}^1 := \mathbb{1}_{x_1}(X)$, $P^1(\mathbf{C}_1) = P(Z \mid X = x_2)$ with $\mathbf{C}_1 = \{Z\}$ and $\mathbf{S}_0^b = \{X\}$, $P^2(\mathbf{C}_2 \mid \mathbf{S}_1) = P(W \mid X, Z)$ with $\mathbf{C}_2 = \{W\}$ and $\mathbf{S}_1 = \{X, Z\}$, $P^3(Y \mid \mathbf{S}_2) = P(Y \mid X = x_0, W, Z)$ with $\mathbf{S}_2 = \{W, Z\}$ and $\mathbf{S}_2^b = \{X\}$.*

Despite the broad expressiveness of UCA, as illustrated in this section and ppendix B, not all causal estimand functionals are UCA-expressible. To witness, consider the 'napkin' estimand described in (Pearl and Mackenzie, 2018; Jung et al., 2021a) with $\mathcal{G}$ in Fig. 1c, defined as $P(y \mid \mathrm{do}(x)) = \frac{\sum_w P(y,x \mid r,w) P(w)}{\sum_w P(x \mid r,w) P(w)}$. Here, the functional for $\mathbb{E}[Y \mid \mathrm{do}(x)]$ is represented not as the expectation of a product of conditional distributions, but rather as a quotient of sums of conditional distributions. The napkin estimand is not UCA-expressible. Intuitively, if a target functional is expressed as

an expectation of a probability measure that is represented as a product of multiple conditional distributions, it can be captured through UCA. A formal criterion is the following:

**Theorem 1** (**Expressiveness**). *Suppose a functional $\psi_0$ is expressed as the mean of the following measure, $P^{m+1}(Y \mid \mathbf{S}'_m) \prod_{i=1}^m P^i(\mathbf{C}_i \mid \mathbf{S}'_{i-1}) \sigma^i_{\mathbf{R}_i}(\mathbf{R}_i \mid \mathbf{S}'_i \setminus \mathbf{R}_i)$, where $\mathbf{S}'_i = (\mathbf{C}^{(i)} \cup \mathbf{R}^{(i)}) \setminus \mathbf{S}^b_i$ for each $i = 1, \ldots, m$ and $P^j(\mathbf{V})$ for $j = 1, \ldots, m+1$ are distributions of the form $P^j(\mathbf{V}) = Q^j(\mathbf{V} \mid \mathbf{S}^b_{j-1} = \mathbf{s}^b_{j-1})$. Then, the functional $\psi_0$ can be expressed through UCA in Eq. (1).*

## 3 Scalable Estimator for Unified Covariate Adjustment

So far, we discussed the *coverage* of UCA. In this section, we construct a *scalable* estimator for UCA that achieves doubly robustness property and provides its finite sample guarantee. We define the estimator with two sets of nuisance parameters $\boldsymbol{\mu}$ and $\boldsymbol{\pi}$. $\boldsymbol{\mu}$ is a collection of regression parameters, and $\boldsymbol{\pi}$ is a collection of ratio parameters.

We introduce sets to define regression nuisances. Define $\mathbf{B}_{i-1} := \mathbf{S}_i \cap \mathbf{C}^{(i-1)} \cap \mathbf{S}^b_{i-1}$ for $i = 2, \cdots, m$ as a bifurcated set, which is a subset of $\mathbf{S}_i$ in $P^{i+1}(\mathbf{C}_{i+1} \mid \mathbf{S}_i)$ that is fixed to $\mathbf{s}^b_{i-1}$ at $P^i(\mathbf{C}_i \mid \mathbf{S}_{i-1})$, while marginalized out over $P^j(\mathbf{C}_j \mid \mathbf{S}_{j-1})$ for some $j < i$ (e.g., $X$ in FD). Set $\mathbf{B}_m = \emptyset$. We use $\mathbf{B}'_{i-1}$ to denote an independent copy of $\mathbf{B}_{i-1}$ (variables following the same distribution as $\mathbf{B}_{i-1}$ but independent of $\mathbf{B}_{i-1}$ and $\mathbf{V}$). With $\mathbf{B}_{i-1}$ and $\mathbf{B}'_{i-1}$, we define $\mathbf{S}'_i := ((\mathbf{S}_i \cup \mathbf{B}_i) \setminus \mathbf{B}_{i-1}) \cup \mathbf{B}'_{i-1}$ and $\check{\mathbf{S}}_i := \mathbf{S}'_i \setminus \mathbf{R}_i$ for $i = 2, \cdots, m$. Define the regression nuisance parameters as follows: $\mu^m_0(\mathbf{S}_m) := \mathbb{E}_{P^{m+1}}[Y \mid \mathbf{S}_m]$ and $\check{\mu}^m_0(\check{\mathbf{S}}_m) := \sum_{\mathbf{r}_m} \sigma^m_{\mathbf{R}_m}(\mathbf{r}_m \mid \mathbf{S}_m \setminus \mathbf{R}_m) \mu^m_0(\mathbf{r}_m, \check{\mathbf{S}}_m)$. For $i = m-1, \cdots, 1$,

$$\mu^i_0(\mathbf{S}_i, \mathbf{B}'_i) := \mathbb{E}_{P^{i+1}}[\check{\mu}^{i+1}_0(\check{\mathbf{S}}_{i+1}) \mid \mathbf{S}_i, \mathbf{B}'_i], \tag{6}$$

$$\check{\mu}^i_0(\check{\mathbf{S}}_i) := \sum_{\mathbf{r}_i} \sigma^i_{\mathbf{R}_i}(\mathbf{r}_i \mid \mathbf{S}_i \setminus \mathbf{R}_i) \mu^i_0(\mathbf{r}_i, \mathbf{S}'_i). \tag{7}$$

Equipped with the regression nuisances, UCA can be computed as follows:

**Proposition 2.** *UCA in Eq. (1) can be parameterized as $\psi_0 = \mathbb{E}_{P^1}[\check{\mu}^1_0(\check{\mathbf{S}}_1)]$.*

Whenever no variables are being summed and fixed simultaneously (i.e., $\mathbf{B}_{i-1} = \emptyset$ for all $i = 2, \cdots, m$) in the UCA functional, as in Eq. (5) in Ctf-DE, the standard SBD adjustment or examples in Appendix B, we can estimate $\boldsymbol{\mu}$ through nested regression methods with off-the-shelf regression models and compute UCA in Eq. (1) as $\psi_0 = \mathbb{E}_{P^1}[\check{\mu}^1_0(\check{\mathbf{S}}_1)]$. This approach aligns with existing SBD estimators (Bang and Robins, 2005; Robins et al., 2009; van der Laan and Gruber, 2012; Rotnitzky et al., 2017; Luedtke et al., 2017; Díaz et al., 2023). For instance, in Ctf-DE in Example 3, $\mu^2_0(W, Z) := \mathbb{E}_P[Y \mid W, Z, x_0]$, $\check{\mu}^2_0(W, Z) = \mu^2_0(W, Z)$, $\mu^1_0(X, Z) := \mathbb{E}_P[\check{\mu}^2_0(W, Z) \mid X, Z]$, $\check{\mu}^1_0(Z) = \mu^1_0(x_1, Z)$, and $\psi_0 = \mathbb{E}_P[\check{\mu}^1_0(Z) \mid x_2]$. These nuisances can be estimated efficiently with regression models run in polynomial time relative to the number of variables and samples (e.g., neural networks (LeCun et al., 2015) or XGBoost (Chen and Guestrin, 2016)).

Beyond the SBD framework, the regression nuisances are capable of representing functionals in the presence of variables being summed and fixed simultaneously (e.g., FD in Eq. (2) or Verma in Eq. (3)). As an example, consider FD in Eq. (2) with its UCA representation in Example 1. First, define $\mu^2_0(Z, X, C) := \mathbb{E}_P[Y \mid Z, X, C]$ with $\mathbf{S}_2 = \{Z, X, C\}$. Next, we have $\mathbf{B}_1 = \mathbf{S}^b_1 \cap \mathbf{C}_1 = \{X\}$ and, $\check{\mathbf{S}}_2 = \{Z, X', C\}$, where $X'$ is an independent copy of $X$. Consequently, $\check{\mu}^2_0(Z, X', C) := \mu^2_0(Z, X', C)$, where $(Z, X', C)$ is plugged into a function $\mu^2_0$. Next, define $\mu^1_0(C, X') := \mathbb{E}_P[\check{\mu}^2_0(Z, X', C) \mid x, C, X']$. Finally, we have $\check{\mu}^1_0(C, X) = \mu^1_0(C, X)$. The expectation, $\mathbb{E}_P[\check{\mu}^1_0(C, X)] = \sum_{c, x'} P(c, x') \check{\mu}^1_0(c, x')$, correctly specifies FD in Eq. (2) as follows:

$$\sum_{c, x'} P(c, x') \check{\mu}^1_0(c, x') = \sum_{c, x'} P(c, x') \mu^1_0(c, x') = \sum_{c, x'} P(c, x') \mathbb{E}_P[\check{\mu}^2_0(Z, x', c) \mid x, c, x']$$

$$=^* \sum_{c, x', z} P(c, x') P(z \mid x, c) \mu^2_0(z, x', c) = \sum_{c, x', z} P(c, x') P(z \mid x, c) \mathbb{E}_P[Y \mid z, x', c],$$

where the equation $=^*$ holds since $X'$ is an independent copy of $X$, so it's independent of $Z$.

Empirically, generating $\mathbf{B}'_i$ involves permuting copied samples of $\mathbf{B}_i$, an used in recent works in (Chernozhukov et al., 2022; Xu and Gretton, 2022). We name this approach *empirical bifurcation*:

**Algorithm 2:** DML-UCA($\{\mathcal{D}^i\}, L$)

1 (**Sample splitting**) For each $i \in [m+1]$, randomly split $\mathcal{D}^i \overset{iid}{\sim} P^i$ into $L$-folds. Let $\mathcal{D}^i_\ell$ denote the $\ell$-th partition, and define $\mathcal{D}^i_{-\ell} := \mathcal{D}^i \setminus \mathcal{D}^i_\ell$. We use $\mathbf{W}(\mathcal{D}^i_\ell)$ to refer to the samples of $\forall \mathbf{W}$ in $\mathcal{D}^i_\ell$.

2 **for** $\ell \in [L]$ **do**

3      **for** $i = m, \ldots, 1$ **do**

         1. Learn $\hat{\mu}^i_{-\ell}(\mathbf{S}_i, \mathbf{B}'_i)$ by regressing $\check{\mu}^{i+1}_{-\ell}(\check{\mathbf{S}}_{i+1}(\mathcal{D}^{i+1}_{-\ell}))$ onto $\mathbf{S}_i(\mathcal{D}^{i+1}_{-\ell}), \mathbf{B}'_i(\mathcal{D}^{i+1}_{-\ell})$ (where $\check{\mu}^{m+1} := Y$).

         2. Evaluate $\check{\mu}^i_{-\ell}(\check{\mathbf{S}}_i(\mathcal{D}^i_{-\ell}))$ using empirical bifurcation under the policy $\sigma^i_{\mathbf{R}_i}$.

         3. Compute $\check{\mu}^i_\ell := \check{\mu}^i_{-\ell}(\check{\mathbf{S}}_i(\mathcal{D}^i_\ell))$ by evaluating $\check{\mu}^i_{-\ell}$ using samples $\mathcal{D}^i_\ell$.

         4. For a nuisance parameter $\pi^i_0$ satisfying Eq. (9), learn $\hat{\pi}^i_{-\ell}$ using samples $\{\mathcal{D}^j_{-\ell} : j \in [i+1]\}$.

         5. Evaluate $\hat{\pi}^i_\ell := \hat{\pi}^i_{-\ell}(\{\mathcal{D}^j_\ell : j \in [i+1]\})$.

4      **end**

5 **end**

6 Return the DML-UCA estimator $\hat{\psi}$:

$$\hat{\psi} := \frac{1}{L} \sum_{\ell=1}^{L} \sum_{i=1}^{m} \mathbb{E}_{\mathcal{D}^{i+1}_\ell}[\hat{\pi}^i_\ell(\check{\mu}^{i+1}_\ell - \hat{\mu}^i_\ell)] + \mathbb{E}_{\mathcal{D}^1_\ell}[\check{\mu}^1_\ell]. \tag{8}$$

**Definition 2** (**Empirical bifurcation**). *An **empirical bifurcation** for $\mathbf{B}$ following a distribution $P$ is the procedure of copying samples of $\mathbf{B} \sim P$ and randomly permuting to obtain new samples $\mathbf{B}'$.*

In general, the regression nuisances can be estimated from data by employing empirical bifurcation and off-the-shelf regression models.

Next, we define the ratio nuisance parameters $\boldsymbol{\pi}$. Define $\pi^m_0(\mathbf{S}_m)$ as the solution functional satisfying $\mathbb{E}_{P^{m+1}}[\mu^m_0(\mathbf{S}_m)\pi^m_0] = \psi_0$. Recursively, for $i = m-1, \cdots, 1$, define $\pi^i_0(\mathbf{S}_i, \mathbf{B}'_i)$ as a functional satisfying the following equation, for any $\mu^{i+1} \in L_2(P^{i+2})$.

$$\mathbb{E}_{P^{i+2}}[\pi^{i+1}_0(\mathbf{S}_{i+1}, \mathbf{B}'_{i+1})\mu^{i+1}(\mathbf{S}_{i+1}, \mathbf{B}'_{i+1})] = \mathbb{E}_{P^{i+1}}[\pi^i_0(\mathbf{S}_i, \mathbf{B}'_i)\mathbb{E}_{P^{i+1}}[\check{\mu}^{i+1}(\check{\mathbf{S}}_{i+1}) \mid \mathbf{S}_i, \mathbf{B}'_i]], \tag{9}$$

where the closed form solution is given as follows:

$$\pi^i_0 = \frac{\prod_{j=1}^{i} P^j(\mathbf{C}_j \mid \mathbf{S}_j)\sigma^j_{\mathbf{R}_j}(\mathbf{R}_j \mid \mathbf{S}_j \setminus \mathbf{R}_j)}{P^{i+1}(\mathbf{S}_i, \mathbf{B}'_i)} \tag{10}$$

For the example of FD, $\pi^2_0 = \frac{P(Z|x,C)}{P(Z|X,C)}$ and $\pi^1_0 = \frac{P(x)}{P(x|C)}$.

Equipped with the ratio nuisances, UCA can be computed as follows:

**Proposition 3.** *UCA in Eq. (1) can be parameterized as $\psi_0 = \mathbb{E}_{P^{m+1}}[\pi^m_0 Y]$.*

Estimating the ratio nuisances may be challenging due to the distribution ratio of continuous/high-dimensional variables. To address the challenge, we use Bayes' rule to transform the distribution ratio into a more tractable form. For example, in FD, if the treatment $X$ is a singleton binary, instead of estimating $\pi^2_0 = \frac{P(Z|x,C)}{P(Z|X,C)}$, an equivalent estimand $\pi^2_0 = \frac{P(x|Z,C)P(X|C)}{P(X|Z,C)P(x|C)}$ can be estimated. This approach allows to use off-the-shelf probabilistic classification methods for estimating distribution ratios, allowing scalable computation. A detailed procedure for ratio estimation is in Appendix C.2.

Combining regression and ratio-nuisances, we present a double/debiased machine learning (DML) (Chernozhukov et al., 2018)-based estimator $\hat{\psi}$ for the UCA, titled 'DML-UCA', in Algo. 2. We provide detailed nuisance specification for various examples in Appendix A and B.

DML-UCA provides a scalable estimator for functionals expressible through UCA. When the target query is BD/SBD, DML-UCA aligns with existing doubly robust SBD estimators (Bang and Robins, 2005; Robins et al., 2009; van der Laan and Gruber, 2012; Rotnitzky et al., 2017; Luedtke et al., 2017; Díaz et al., 2023). Beyond SBD, DML-UCA can be estimated in polynomial time relative to the number of variables and samples, ensuring its scalability:

**Theorem 2** (**Scalability**). *Algo. 2 runs in $O(Kn_{\max} + T(m, n_{\max}, K))$, where $K$ is the number of distinct in $\mathbf{P}$, $n_{\max} := \max\{|\mathcal{D}^k| : k \in [K]\}$, and $T(m, n_{\max}, K)$ is the time complexity for*

| Estimand | Estimator | Complexity |
|---|---|---|
| | Plug-in | $O(n2^m)$ |
| BD/SBD | IPW (Rosenbaum and Rubin, 1983) OM (Robins, 1986) AIPW (Rotnitzky et al., 1998) | $O(n + T(m, n))$ |
| FD Tian's | Fulcher et al. (2019); Guo et al. (2023) Bhattacharya et al. (2022) | $O(n2^m + T(m, n))$ |
| **UCA** | DML-UCA (BD, FD and Tian's) | $O(n + T(m, n))$ |
| | DML-UCA (general) | $O(Kn_{\max} + T(m, n_{\max}, K))$ |
| obsID | DML-ID (Jung et al., 2021a) | $O(n2^{2m} + T(m, n))$ |
| gID | DML-gID (Jung et al., 2023a) | $O(Kn_{\max}2^{2m} + T(m, n_{\max}, K))$ |

Table 2: Comparison of time complexities of existing estimators for estimands: $n_{\max} := \max\{|\mathcal{D}^i|\}$ is the number of samples, $m$ is the number of variables, and $T(m, n_{\max}, K)$ (or $T(m, n) := T(m, n_{\max} = n, K = 1)$) is the time complexity for learning nuisance parameters for the target functional. The plug-in estimator for BD is one where $\mathbb{E}_P[Y \mid \mathbf{x}, \mathbf{z}]$ and $P(\mathbf{z})$ are estimated from data, and $\sum_{\mathbf{z}} \mathbb{E}_P[Y \mid \mathbf{x}, \mathbf{z}]P(\mathbf{z})$ is evaluated. Details are in Sec. C.4.

*learning nuisances $\hat{\mu}_\ell^i$ and $\hat{\pi}_\ell^i$. Specifically, $O(T(m, n_{\max}, K)) = O(K \times L \times (T_{\boldsymbol{\mu}} + T_{\boldsymbol{\pi}}))$, where $T_{\boldsymbol{\mu}} := \max\{T_{\hat{\mu}_\ell^i} : i \in [m], \ell \in [L]\}$, $T_{\boldsymbol{\pi}} := \max\{T_{\hat{\pi}_\ell^i} : i \in [m], \ell \in [L]\}$, and $T_{\hat{\mu}_\ell^i}$ and $T_{\hat{\pi}_\ell^i}$ denote the time complexity for learning and evaluating $\hat{\pi}_\ell^i$ and $\hat{\mu}_\ell^i$, respectively.*

An an example, for XGBoost (Chen and Guestrin, 2016), $T_{\boldsymbol{\pi}} = T_{\boldsymbol{\mu}} = O(\text{num}_{\text{tree}} \times \text{depth}_{\text{tree}} \times n_{\max} \log n_{\max})$, where $\text{num}_{\text{tree}}$ and $\text{depth}_{\text{tree}}$ are the number and depth of trees in XGBoost.

Table 2 summarizes the comparison of time complexities for existing estimators. As shown in the table, scalable estimators with polynomial time complexity have only been developed for BD/SBD estimands. Existing estimators beyond SBD often lack scalability. For instance, existing estimators for FD (Fulcher et al., 2019; Guo et al., 2023) or Tian's adjustment (Bhattacharya et al., 2022) face exponential time complexity in the dimension of mediators. In contrast, DML-UCA's polynomial time complexity positions it as a uniquely scalable solution within the UCA functional class, which includes FD and Tian's adjustment as special cases. For general obsID/gID estimands beyond the UCA class, scalable estimators have yet to be developed.

## 3.1 Error analysis

In this section, we show that DML-UCA exhibits doubly robustness, in addition to scalability. Since UCA is composed of multiple (possibly distinct) distributions, we provide a tool to distinguish them.

**Definition 3** (**Index set**). *The index sets $\mathcal{I}_1, \cdots, \mathcal{I}_K$ partition $\{1, \cdots, m+1\}$ such that indices $i$ and $j$ are in the same set $\mathcal{I}_k$ if and only if $P^i(\mathbf{V}) = P^j(\mathbf{V})$.*

We will use $\mathsf{P}^k$ for $k = 1, \cdots, K$ to denote the distribution $P^i$ for $i \in \mathcal{I}_k$. Then, the functional $\Psi[\mathbf{P}; \boldsymbol{\sigma}]$ in Eq. (1) can be written as follows:

$$\Psi[\mathbf{P}; \boldsymbol{\sigma}] = \Psi[\{\mathsf{P}^k : k = 1, \cdots, K\}; \boldsymbol{\sigma}]. \tag{11}$$

Since multiple distributions are involved in UCA, deriving an influence function for each distribution $\mathsf{P}^k$ becomes necessary. A standard influence function is typically defined for a single distribution $P$, and thus, does not suffice for studying multi-distribution setting. To address the issue, we employ a *partial influence function* (PIF) (Pires and Branco, 2002), an influence function defined relative to each $\mathsf{P}^k$. A formal definition is in Appendix C. For UCA, PIFs are given as follows:

**Theorem 3** (**PIF for UCA**). *Assume that $\mu_0^i < \infty$ and $0 < \pi_0^i < \infty$ almost surely for $i = 1, \cdots, m$. Define $\eta_0^1 := \{\mu_0^1\}$ and $\eta_0^i := \{\pi_0^{i-1}, \mu_0^i, \mu_0^{i-1}\}$ for $i = 1, \cdots, m+1$, and*

$$\varphi^i(\check{\mathbf{S}}_i; \eta_0^i, \psi_0) := \begin{cases} \pi_0^{i-1}\{\check{\mu}_0^i - \mu_0^{i-1}\} & \text{if } i > 1 \\ \check{\mu}_0^1 - \psi_0 & \text{if } i = 1. \end{cases} \tag{12}$$

*Let* $\mathbf{V}^k := \cup_{i \in \mathcal{I}_k} \check{\mathbf{S}}^i$ *and* $\boldsymbol{\eta}_0^k := \cup_{i \in \mathcal{I}_k} \eta_0^i$. *Then, the* $k$-*th PIF for UCA is* $\phi_0^k := \phi^k(\mathbf{V}^k; \boldsymbol{\eta}_0^k, \psi_0) := \sum_{i \in \mathcal{I}_k} \varphi^i(\mathbf{S}^i; \eta_0^i, \psi_0)$.

Equipped with PIFs, we provide a finite-sample guarantee for DML-UCA, extending Chernozhukov et al. (2023) which analyzed DML estimators for BDs.

**Theorem 4 (Finite sample guarantee).** *Suppose* $\mu_0^i, \hat{\mu}_\ell^i < \infty$ *and* $0 < \pi_0^i, \hat{\pi}_\ell^i < \infty$ *almost surely for* $i = 1, \cdots, m$. *Suppose the third moment of* $\phi_0^k$ *for* $k = 1, \cdots, K$ *exist. Let* $\phi_0^k := \phi^k(\mathbf{V}^k; \boldsymbol{\eta}_0^k, \psi_0)$ *and* $\hat{\phi}_\ell^k := \phi^k(\mathbf{V}^k; \hat{\boldsymbol{\eta}}_\ell^k, \psi_0)$. *Let* $R_1^k := (1/L) \sum_{\ell=1}^L (\mathbb{E}_{\mathcal{D}_\ell^k}[\hat{\phi}_\ell^k] - \mathbb{E}_{\mathsf{P}^k}[\hat{\phi}_\ell^k])$. *Then,*

1. *The error* $\hat{\psi} - \psi_0$ *is decomposed as follows:*

$$\hat{\psi} - \psi_0 = \sum_{k=1}^K R_1^k + \frac{1}{L} \sum_{\ell=1}^L \sum_{i=1}^m \mathbb{E}_{P^{i+1}}[(\hat{\pi}_\ell^i - \pi_0^i)(\mu_0^i - \hat{\mu}_\ell^i)]. \quad (13)$$

2. *Let* $\rho_{k,0}^2 := \mathbb{V}_{\mathsf{P}^k}[\phi_0^k]$. *With probability (W.P) greater than* $1 - \epsilon$,

$$\sum_{k=1}^K R_1^k \le K \sqrt{\frac{2}{\epsilon}} \left( \sqrt{\sum_{k=1}^K \frac{\rho_{k,0}^2}{|\mathcal{D}^k|}} + \sqrt{\sum_{\ell=1}^L \sum_{k=1}^K \frac{\|\hat{\phi}_\ell^k - \phi_0^k\|_{\mathsf{P}^k}^2}{|\mathcal{D}_\ell^k|}} \right). \quad (14)$$

3. *Let* $\kappa_{k,0}^3 := \mathbb{E}_{\mathsf{P}^k}[|\phi_0^k|^3]$. *Let* $\mathtt{NORMAL}(x)$ *denote the standard normal CDF. W.P greater than* $1 - \epsilon$,

$$\left| \mathsf{P}^k \left( \frac{\sqrt{|\mathcal{D}^k|}}{\rho_{k,0}} R_1^k < x \right) - \mathtt{NORMAL}(x) \right| \le \frac{1}{\sqrt{2\pi}} \sqrt{\frac{1}{\epsilon} \sum_{\ell=1}^L \frac{\|\hat{\phi}_\ell^k - \phi_0^k\|_{\mathsf{P}^k}^2}{|\mathcal{D}_\ell^k|}} + \frac{0.4748 \kappa_{k,0}^3}{\rho_{k,0}^3 \sqrt{|\mathcal{D}^k|}}, \quad (15)$$

This is a novel finite sample guarantee of DML-based estimators for functionals beyond SBD. Finite sample analyses for functionals beyond SBD have been studied only for the non-doubly robust estimators (Bhattacharyya et al., 2022). For doubly robust estimators, only asymptotic analyses were provided for FD (Fulcher et al., 2019; Guo et al., 2023), Tian's adjustment (Bhattacharya et al., 2022), and obsID (Jung et al., 2021b). Thm. 4 elucidates that the error can be decomposed into two terms $R_1^k$ and $R_2^\ell$. The term $R_1^k$ closely approximates a standard normal distribution variable, and $R_2^\ell$, comprises the error of $(\hat{\pi}_\ell^i, \hat{\pi}_\ell^{i-1})$ and $\hat{\mu}^i$, exhibiting doubly-robustness behavior. Specifically, if the nuisance parameters $\hat{\mu}_\ell^i, \hat{\pi}_\ell^i$, and $\hat{\pi}_\ell^{i-1}$ converge at a rate of $n^{-1/4}$ (where $n$ represents the size of the smallest sample set), then DML-UCA converges at a faster rate of $n^{-1/2}$. This point becomes evident in the corresponding asymptotic analysis:

**Corollary 4 (Asymptotic error).** *Assume* $\mu_0^i, \hat{\mu}_0^i < \infty$ *and* $0 < \pi_0^i, \hat{\pi}_0^i < \infty$ *almost surely. Suppose the map* $\hat{\boldsymbol{\eta}}_\ell^k \mapsto \hat{\phi}_\ell^k$ *is uniformly differentiable with respect to* $\hat{\boldsymbol{\eta}}_\ell^k$, *and the derivative of* $\hat{\phi}_\ell^k$ *w.r.t.* $\hat{\boldsymbol{\eta}}_\ell^k$ *is bounded by some constants. Suppose* $\hat{\mu}_\ell^i$ *and* $\hat{\pi}_\ell^i$ *are* $L_2$-*consistent. Then,*

$$\hat{\psi} - \psi_0 = \sum_{k=1}^K R_1^k + \frac{1}{L} \sum_{\ell=1}^L \sum_{i=1}^m O_{P^{i+1}} \left( \|\hat{\mu}_\ell^i - \mu_0^i\|(\|\hat{\pi}_\ell^i - \pi_0^i\|) \right),$$

*and* $\sqrt{|\mathcal{D}^k|} R_1^k$ *converges in distribution to* $normal(0, \rho_{k,0}^2)$.

## 4 Experiments

In this section, we demonstrate the *scalability* and *doubly robustness* of the DML-UCA estimator, where nuisances are learned through XGBoost (Chen and Guestrin, 2016). We specify an SCM $\mathcal{M}$ for FD (Fig. 1a), Verma (Fig. 1b), and the example graph in (Jung et al., 2021a) (Fig. 1e), and generate datasets $\mathsf{D}^k \sim \mathsf{P}^k$ from the SCM. The target estimand is denoted as $\psi_0$. Details are in Appendix F. Further simulations are provided in Appendix E.

**Scalability.** To demonstrate scalability of DML-UCA, we compare the running time with existing estimators of (Fulcher et al., 2019) (FD) and (Jung et al., 2021a) (Verma's equation and the estimand

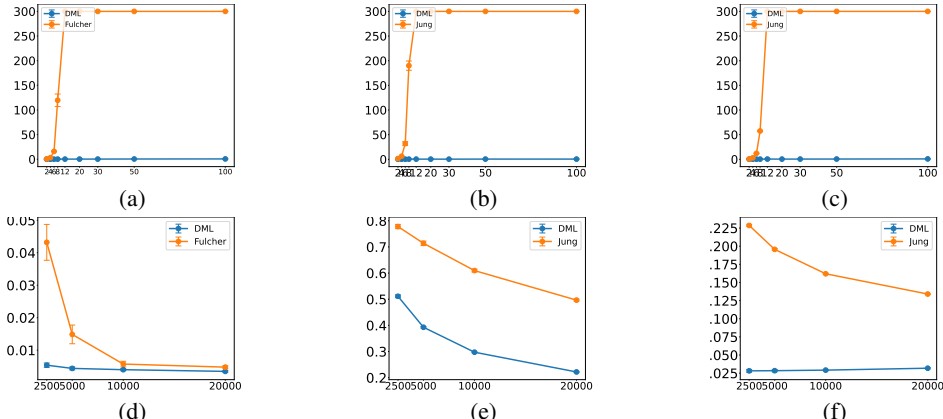

Figure 2: Comparison of DML-UCA ('DML') with existing estimators using **(Top)** running-time-plots (*x-axis*: the dimension of summed variables, *y-axis*: running time); and **(Bottom)** AAE-plots (*x-axis*: the sample size, *y-axis*: errors). DML-UCA is compared with **(a,d)** Fulcher et al. (2019) for FD; **(b,e)** (Jung et al., 2021a) for Verma's equation; and **(c,f)** Jung et al. (2021a) for Jung's equation.

with Fig. 1e — $\mathbb{E}[Y \mid \mathrm{do}(x_1, x_2)] = \sum_{x_1', r, z} \mathbb{E}_P[Y \mid r, x_1', x_2, z] P(r \mid x_1, z) P(z, x_1')$ — which we call 'Jung's equation'). For each example, we increment the dimension of the summed variables, run 100 simulations, take the average of running times, and compare this average. We label this plot as 'run-time-plot', presented in the top side of Fig. 2. In the comparison with (Fulcher et al., 2019) for FD in Fig. 2a, we fix $|C| = 2$ and increment $|Z| = \{2, 4, 6, 8, 12, 20, 30, 50, 100\}$. When comparing with (Jung et al., 2021a), for Verma's equations in Figs. (2b), we fix $|A| = 2$ and increment $|B| = \{2, 4, 6, 8, 12, 20, 30, 50, 100\}$. For Jung's equation in Fig. 2c, we fix $|Z| = 2$ and $|R| = \{2, 4, 6, 8, 12, 20, 30, 100\}$. The timeout for the run-time is set to 300 seconds. For all scenarios, the run-time of existing estimators increases rapidly over dimensions due to the summation operation while DML-UCA scales well for high-dimensional covariates.

**Doubly robustness.** To demonstrate doubly robustness, we compare the error of DML-UCA with existing estimators for FD of Fulcher et al. (2019) and for Verma's and Jung's equations of Jung et al. (2021a) We use $\hat{\psi}^{\mathrm{est}}$ for est $\in \{\mathrm{DML}, \mathrm{Fulcher}, \mathrm{Jung}\}$ to denote each estimator. We use the average absolute error (AAE), which is an average of the error of the estimated versus true causal effect of $\mathbf{X} = \mathbf{x}$: $\frac{1}{|\mathrm{domain}(\mathbf{X})|} \sum_{\mathbf{x} \in \mathrm{domain}(\mathbf{X})} |\hat{\psi}^{\mathrm{est}}(\mathbf{x}) - \psi_0(\mathbf{x})|$. To witness the fast convergence of DML-UCA, we enforce the convergence rate of nuisance estimates to be no faster than the decaying rate $n^{-1/4}$ by adding the noise term $\epsilon \sim \texttt{normal}(n^{-1/4}, n^{-1/4})$ to nuisances, inspired by the experimental design in (Kennedy, 2023). We ran 100 simulations for each number of samples $n = \{2500, 5000, 10000, 20000\}$. We label the plot as 'AAE-plot', presented in the bottom side of Fig. 2. For each example, DML-UCA outperforms other estimators, exhibiting fast convergence.

## 5   Conclusions

We introduce a framework that encompasses a broad class of sum-product causal estimands, called UCA class, for which scalable estimators were previously unavailable. We demonstrate the expressiveness of the UCA class, which includes not only BD/SBD but also broader classes such as Tian's adjustment incorporating FD and Verma, and Ctf-DE, for which the existing SBD-based framework is not applicable. We develop an estimator for UCA called DML-UCA that can estimate the target estimand in polynomial time relative to the number of samples and variables, ensuring scalability. We provide finite-sample guarantees and corresponding asymptotic error analysis for DML-UCA, demonstrating its fast convergence. These scalability and fast convergence properties are empirically verified through simulations. Our results pave the way toward developing an estimation framework maximizing both coverage and scalability in Table 1.

## Acknowledgments

We thank anonymous reviewers for constructive comments to improve the manuscript. This research is supported in part by the NSF, ONR, AFOSR, DoE, Amazon, JP Morgan, and The Alfred P. Sloan Foundation.

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

# Supplement to "Unified Covariate Adjustment for Causal Inference"

## Contents

# A Nuisance Specification

## A.1 Front-door adjustment in Example 1

We first note that the front-door adjustment is an expectation of the following product:

$$P(C, X)P(Z \mid x, C)P(Y \mid C, X, Z),$$

which implies that

- $\mathbf{C}_1 = \{C, X\}$; $\mathbf{C}_2 = \{Z\}$; $\mathbf{C}_3 = \{Y\}$ (i.e., $m = 2$).
- $\mathbf{S}_1 = \{C\}$ and $\mathbf{S}_2 = \{C, X, Z\}$
- $\mathbf{S}_1^b = \{X\}$
- $\mathbf{B}_1 = \mathbf{S}_2 \cap \mathbf{C}_1 \cap \mathbf{S}_1^b = \{X\}$.
- $\check{\mathbf{S}}_2 = \{C, X', Z\}$ and $\check{\mathbf{S}}_1 = \{C, X\}$.
- $P^2 = P(\cdot \mid x)$.

The regression nuisances are the followings:

$$\mu_0^2(\mathbf{S}_2) := \mu_0^2(Z, X, C) := \mathbb{E}_P[Y \mid Z, X, C]$$
$$\check{\mu}_0^2(\check{\mathbf{S}}_2) := \mu_0^2(Z, X', C)$$
$$\mu_0^1(\mathbf{S}_1, \mathbf{B}_1') := \mu_0^1(X', C) := \mathbb{E}_P[\mu_0^2(Z, X', C) \mid x, C, X']$$
$$\check{\mu}_0^1(\check{\mathbf{S}}_1) = \mu_0^1(X, C).$$

The ratio nuisances are the followings:

$$\pi_0^2(Z, X, C) = \frac{P(Z \mid x, C)}{P(Z \mid X, C)}, \tag{A.1}$$

$$\pi_0^1(C) = \frac{P(x)}{P(x \mid C)}. \tag{A.2}$$

The representation for DML-UCA is

$$\mathbb{E}_P[\pi_0^2(Z, X, C)\{Y - \mu_0^2(Z, X, C)\}]$$
$$+ \mathbb{E}_P[\pi_0^1(C)\{\mu_0^2(Z, X', C) - \mu_0^1(X', C)\} \mid x]$$
$$+ \mathbb{E}_P[\mu_0^1(X, C)].$$

## A.2 Verma's equation in Example 2

We first note that the Verma's equation in Eq. (3) is an expectation of the following product:

$$P(X)P(A \mid x)P(B \mid A, X)P(Y \mid B, A, x),$$

which implies that

- $\mathbf{C}_1 = \{X\}$; $\mathbf{C}_2 = \{A\}$; $\mathbf{C}_3 = \{B\}$; and $\mathbf{C}_4 = \{Y\}$ (i.e., $m = 3$).
- $\mathbf{S}_1 = \emptyset$; $\mathbf{S}_2 = \{A, X\}$ and $\mathbf{S}_3 = \{B, A\}$.
- $\mathbf{S}_1^b = \{X\}$ and $\mathbf{S}_3^b = \{X\}$.
- $\mathbf{B}_1 = \mathbf{S}_2 \cap \mathbf{C}_1 \cap \mathbf{S}_1^b = \{X\}$.
- $\check{\mathbf{S}}_3 = \{B, A\}$; $\check{\mathbf{S}}_2 = \{A, X'\}$ and $\check{\mathbf{S}}_1 = \{X\}$.
- $P^2 = P^4 = P(\cdot \mid x)$.

The regression nuisances are the followings:

$$\mu_0^3(\mathbf{S}_3) := \mu_0^3(B, A) := \mathbb{E}_P[Y \mid B, A, x]$$
$$\check{\mu}_0^3(\check{\mathbf{S}}_3) := \mu_0^3(B, A) = \mathbb{E}_P[Y \mid B, A, x]$$
$$\mu_0^2(\mathbf{S}_2) := \mu_0^2(A, X) := \mathbb{E}_P[\mu_0^3(B, A) \mid A, X]$$
$$\check{\mu}_0^2(\check{\mathbf{S}}_2) = \mu_0^2(A, X')$$
$$\mu_0^1(\mathbf{S}_1, \mathbf{B}_1') := \mathbb{E}_P[\mu_0^2(A, X') \mid x, X']$$
$$\check{\mu}_0^1(\check{\mathbf{S}}_1) := \mu_0^1(X).$$

The ratio nuisances are the followings:

$$\pi_0^3(B, A) = \frac{\sum_{x'} P(B \mid A, x')P(x')}{P(B \mid A, x)}, \tag{A.3}$$

$$\pi_0^2(A, X) = \frac{P(A \mid x)}{P(A \mid X)} \tag{A.4}$$

$$\tag{A.5}$$

The representation for DML-UCA is

$$\mathbb{E}_P[\pi_0^3(B, A, X')\{Y - \mu_0^3(B, A, x)\} \mid x]$$
$$+ \mathbb{E}_P[\pi_0^2(A, X)\{\mu_0^3(B, A, x) - \mu_0^2(A, X)\}]$$
$$+ \mathbb{E}_P[\check{\mu}_0^2(A, X')].$$

## A.3 Counterfactual directed effect in Example 3

From the fact that Ctf-DE in Eq. (5) is represented as the expectation of $Y$ over $P(Y \mid X = x_0, W, Z)P(W \mid X, Z)P(Z \mid X = x_2)\mathbb{1}_{x_1}(X)$. Set

- $\mathbf{C}_1 = \{Z\}$; $\mathbf{C}_2 = \{W\}$; and $\mathbf{C}_3 = \{Y\}$.

- $\mathbf{S}_1 = \{X, Z\}$, $\mathbf{S}_2 = \{W, Z\}$

- $\mathbf{R}_1 = \{X\}$.

- $\mathbf{S}_0^b = \{X_2\}$; $\mathbf{S}_1^b = \{X\}$; $\mathbf{S}_2^b = \{X\}$.

- $\mathbf{B}_i = \emptyset$ for all $i$.

- $\check{\mathbf{S}}_i = \mathbf{S}_i \setminus \mathbf{R}_i$ for all $i$

The regression nuisances are the followings:

$$\mu_0^2(\mathbf{S}_2) := \mu_0^2(W, Z) := \mathbb{E}_P[Y \mid W, x_0, Z]$$
$$\check{\mu}_0^2(\check{\mathbf{S}}_2) := \mu_0^2(W, Z) = \mathbb{E}_P[Y \mid W, x_0, Z]$$
$$\mu_0^1(\mathbf{S}_1) := \mu_0^1(X, Z) := \mathbb{E}_P[\mu_0^2(W, Z) \mid X, Z]$$
$$\check{\mu}_0^1(\check{\mathbf{S}}_1) = \mu_0^1(x_1, Z).$$

The ratio nuisances are the followings:

$$\pi_0^2(W, Z) = \frac{P(W \mid x_1, Z)P(Z \mid x_2)}{P(W, Z \mid x_0)}, \tag{A.6}$$

$$\pi_0^1(X, Z) = \frac{\mathbb{1}_{x_1}(X)P(Z \mid x_2)}{P(X \mid Z)P(Z)}. \tag{A.7}$$

The representation for DML-UCA is

$$\mathbb{E}_P[\pi_0^2(W, Z)\{Y - \mu_0^2(W, X, Z)\} \mid X = x_0]$$
$$+ \mathbb{E}_P[\pi_0^1(X, Z)\{\mu_0^2(W, x_0, Z) - \mu_0^1(X, Z)\}]$$
$$+ \mathbb{E}_P[\mu_0^1(x_1, Z) \mid x_2].$$

## A.4  Example Estimand for Fig. 1e

Given Fig. 1e, the causal effect is given as

$$\mathbb{E}[Y \mid \mathrm{do}(x_1, x_2)] = \sum_{r,z,x_1'} \mathbb{E}_P[Y \mid r, x_2, z, x_1']P(r \mid x_1, z)P(x_1', z),$$

which is the expectation of $Y$ over the probability measure

$$P(Y \mid R, X_2, Z, X_1)P(R \mid x_1, Z)P(X_1, Z)\mathbb{1}_{x_2}(X_2).$$

Set

- $\mathbf{C}_1 = \{X_1, Z\}$; $\mathbf{C}_2 = \{R\}$; $\mathbf{C}_3 = \{Y\}$

- $\mathbf{R}_1 = \emptyset$ and $\mathbf{R}_2 = \{X_2\}$ with $\sigma_{\mathbf{R}_2}^2 = \mathbb{1}_{x_2}(X_2)$.

- $\mathbf{S}_1 = \{Z\}$, $\mathbf{S}_2 = \{R, X_2, Z, X_1\}$.

- $\mathbf{S}_1^b = \{X_1\}$.

- $\mathbf{B}_1 = \mathbf{S}_2 \cap \mathbf{C}_1 \cap \mathbf{S}_1^b = \{X_1\}$.

- $\check{\mathbf{S}}_2 = \{R, X_1', Z\}$. $\check{\mathbf{S}}_1 = \mathbf{S}_1$.

- $P^2 = P(\cdot \mid x_1)$.

The regression nuisances are the followings:

$$\mu_0^2(\mathbf{S}_2) := \mu_0^2(R, X_2, Z, X_1) := \mathbb{E}_P[Y \mid R, X_2, Z, X_1]$$
$$\check{\mu}_0^2(\check{\mathbf{S}}_2) := \check{\mu}_0^2(R, Z, X_1') = \mathbb{E}_P[Y \mid R, x_2, Z, X_1']$$
$$\mu_0^1(\mathbf{S}_1, \mathbf{B}_1') := \mu_0^1(Z, X_1') := \mathbb{E}_P[\check{\mu}_0^2(R, Z, X_1') \mid x_1, Z, X_1']$$
$$\check{\mu}_0^1(\check{\mathbf{S}}_1) = \mu_0^1(Z, X_1).$$

The ratio nuisances are the followings:

$$\pi_0^2(X_2, R, X_1, Z) = \frac{\mathbb{1}_{x_2}(X_2)}{P(X_2 \mid R, X_1, Z)} \frac{P(R \mid x_1, Z)}{P(R \mid X_1, Z)}, \tag{A.8}$$

$$\pi_0^1(Z) = \frac{P(Z)}{P(Z \mid x)} = \frac{P(x)P(Z)}{P(x \mid Z)P(Z)} = \frac{P(x)}{P(x \mid Z)}. \tag{A.9}$$

The representation for DML-UCA is

$$\mathbb{E}_P[\pi_0^2(X_2, R, X_1, Z)\{Y - \mu_0^2(R, X_2, Z, X_1)\}]$$
$$+ \mathbb{E}_P[\pi_0^1(Z, X')\{\check{\mu}_0^2(R, Z, X_1') - \mu_0^1(Z, X')\} \mid X_1 = x_1]$$
$$+ \mathbb{E}_P[\mu_0^1(Z, X)].$$

# B   More UCA Examples

## B.1   Effect of the treatment on the treated (ETT)

Let $\mathbf{V} = \{\mathbf{Z}, X, Y\}$ be a set of variables where $\mathbf{Z}$ is a covariate, $X$ is a treatment and $Y$ is an outcome. The target estimand is

$$\mathbb{E}[Y(x) \mid x'] = \sum_{\mathbf{z}} \mathbb{E}_P[Y \mid x, \mathbf{z}] P(\mathbf{z} \mid x'). \tag{B.1}$$

The ETT estimand can be written as an expectation of $Y$ over the probability measure

$$\Psi = P(Y \mid X, \mathbf{Z}) P(\mathbf{Z} \mid x') \mathbb{1}_x(X).$$

This factorization implies that $\mathbf{C}_1 := \{\mathbf{Z}\}$, $\mathbf{R} := \mathbf{V} \setminus \mathbf{C}_1 \cup \{Y\} = \{X\}$, where $\mathbf{R}_1 = \{X\}$, and $\sigma_{\mathbf{R}_1}^1 := \mathbb{1}_x(X)$. Also, $\mathbf{S}_1 = \{X\} \cup \mathbf{Z}$. Finally,

$$P^1(\mathbf{C}_1) = P(\mathbf{Z} \mid x')$$
$$P^2(Y \mid \mathbf{S}_1) = P(Y \mid X, \mathbf{Z}).$$

The regression nuisances are the followings:

$$\mu_0^1(\mathbf{S}_1) := \mu_0^1(X, \mathbf{Z}) := \mathbb{E}_P[Y \mid X, \mathbf{Z}]$$
$$\check{\mu}_0^1(\mathbf{S}_1 \setminus \mathbf{R}_1) := \check{\mu}_0^1(\mathbf{Z}) := \mu_0^1(x, \mathbf{Z}).$$

The ratio nuisances are the followings:

$$\pi_0^1(X, \mathbf{Z}) = \frac{P(Z \mid x') \mathbb{1}_x(X)}{P(X, \mathbf{Z})} = \frac{P(x' \mid \mathbf{Z}) P(\mathbf{Z})}{P(x')} \frac{\mathbb{1}_x(X)}{P(X \mid \mathbf{Z}) P(\mathbf{Z})} = \frac{P(x' \mid \mathbf{Z})}{P(X \mid \mathbf{Z})} \frac{\mathbb{1}_x(X)}{P(x')}.$$

The representation for DML-UCA is

$$\mathbb{E}_P[\pi_0^1(X, \mathbf{Z}) \{Y - \mu_0^1(X, \mathbf{Z})\}] + \mathbb{E}_P[\check{\mu}_0^1(\mathbf{Z}) \mid x'].$$

## B.2   Transportability ($S$-admissibility)

Let $\mathbf{V} = \{\mathbf{Z}, X, Y\}$ be a set of variables where $\mathbf{Z}$ is a covariate, $X$ is a treatment and $Y$ is an outcome. Let $S$ denote the domain indicator such that $S = 0$ means the target domain, and $S = 1$ denotes the source. The $S$-admissibility estimand appeared in transportability scenario is

$$\mathbb{E}[Y \mid \mathrm{do}(x)] = \sum_{\mathbf{z}} \mathbb{E}_P[Y \mid x, \mathbf{z}, S = 1] P(\mathbf{z} \mid S = 0). \tag{B.2}$$

The estimand can be written as an expectation of $Y$ over the probability measure

$$\Psi = P(Y \mid X, \mathbf{Z}, S = 1) P(\mathbf{Z} \mid S = 0) \mathbb{1}_x(X).$$

From this factorization, we have $\mathbf{C}_1 := \mathbf{Z}$ and $\mathbf{R}_1 := X$. Also, set $P^1(\mathbf{C}_1) := P(\mathbf{Z} \mid S = 0)$ with $\mathbf{S}_0^b = S$. Set $P^2(Y \mid \mathbf{S}_1) := P(Y \mid X, \mathbf{Z} \mid S = 1)$ with $\mathbf{S}_1^b = S$ and $\mathbf{S}_1 := \{X\} \cup \mathbf{Z}$.

The regression nuisances are the followings:

$$\mu_0^1(\mathbf{S}_1) := \mu_0^1(X, \mathbf{Z}) := \mathbb{E}_P[Y \mid X, \mathbf{Z}, S = 1]$$
$$\check{\mu}_0^1(\check{\mathbf{S}}_1) := \check{\mu}_0^1(\mathbf{Z}) = \mu_0^1(x, \mathbf{Z}).$$

The ratio nuisances are the followings:

$$\pi_0^1(X, \mathbf{Z}) = \frac{\mathbb{1}_x(X)}{P(X \mid \mathbf{Z}, S = 1)} \frac{P(\mathbf{Z} \mid S = 0)}{P(\mathbf{Z} \mid S = 1)}.$$

The representation for DML-UCA is

$$\mathbb{E}_P[\pi_0^1(X, \mathbf{Z})\{Y - \mu_0^1(X, \mathbf{Z})\} \mid S = 1] + \mathbb{E}_P[\check{\mu}_0^1(\mathbf{Z}) \mid S = 0].$$

## B.3 Off-policy evaluation

Let $\mathbf{V} = \{\mathbf{Z}, X, Y\}$ be a set of variables where $\mathbf{Z}$ is a covariate, $X$ is a treatment and $Y$ is an outcome. Let $\sigma^*(X \mid Z)$ denote the behavioral policy that an agent observed; i.e.,

$$(\mathbf{Z}, X, Y) \sim P(Y \mid X, \mathbf{Z})\sigma^*(X \mid \mathbf{Z})P(\mathbf{Z}). \tag{B.3}$$

Let $\sigma(X \mid \mathbf{Z})$ denote a policy to be evaluated. Then, the effect of the policy $\sigma^*$ is given as

$$\mathbb{E}[Y \mid \sigma] := \sum_{x, \mathbf{z}} \mathbb{E}_P[Y \mid x, \mathbf{z}]\sigma^*(x \mid \mathbf{z})P(\mathbf{z}). \tag{B.4}$$

The policy treatment effect in Eq. (B.4) can be represented as UCA as follow.

$$\begin{aligned}
\mathbf{C}_1 &:= \mathbf{Z} \\
\mathbf{R}_1 &:= \{X\} \\
\sigma_{\mathbf{R}_1}^1 &:= \sigma^*(X \mid Z) \\
\mathbf{S}_1 &:= \{X\} \cup \mathbf{Z}.
\end{aligned}$$

Set $P^1(\mathbf{C}_1) \leftarrow P(\mathbf{Z})$, $\sigma_{\mathbf{R}_1}^1(\mathbf{R}_1 \mid \mathbf{Z}_1) \leftarrow \sigma(X \mid Z)$, and $P^2(Y \mid \mathbf{C}_1, \mathbf{R}_1) \leftarrow P(Y \mid X, \mathbf{Z})$. Then,

$$\begin{aligned}
\Psi(\mathbf{P}; \boldsymbol{\sigma}) &:= \sum_{\mathbf{c}, \mathbf{R}} \mathbb{E}_{P^2}[Y \mid \mathbf{c}_1, \mathbf{R}_1]\sigma_{\mathbf{R}_1}^1 P^1(\mathbf{c}_1) \\
&= \sum_{x, \mathbf{z}} \mathbb{E}_P[Y \mid x, \mathbf{z}]\sigma^*(x \mid \mathbf{z})P(\mathbf{z}) \\
&= \mathbb{E}[Y \mid \sigma] \qquad\qquad\qquad \text{(Eq. (B.4))}.
\end{aligned}$$

The regression nuisances are the followings:

$$\begin{aligned}
\mu_0^1(\mathbf{C}^{(1)} \cup \mathbf{R}^{(1)}) &:= \mu_0^1(X, \mathbf{Z}) := \mathbb{E}_P[Y \mid X, \mathbf{Z}] \\
\check{\mu}_0^1(\mathbf{C}^{(1)}) &:= \check{\mu}_0^1(\mathbf{Z}) := \sum_x \mu_0^1(x, \mathbf{Z})\sigma^*(x \mid \mathbf{Z}).
\end{aligned}$$

The ratio nuisances are the followings:

$$\pi_0^1(X, \mathbf{Z}) = \frac{\sigma^*(X \mid \mathbf{Z})}{P(X \mid \mathbf{Z})}.$$

The representation for DML-UCA is

$$\mathbb{E}_P[\pi_0^1(X, \mathbf{Z})\{Y - \mu_0^1(X, \mathbf{Z})\}] + \mathbb{E}_P[\check{\mu}_0^1(\mathbf{Z})].$$

## B.4 Treatment-treatment interactions

Let $\mathbf{V} = \{\mathbf{Z}, X, Y\}$ be a set of variables where $\mathbf{Z}$ is a covariate, $X$ is a treatment and $Y$ is an outcome. The estimand for treatment-treatment interaction discussed in Jung et al. (2023b) is

$$\mathbb{E}[Y \mid \mathrm{do}(x_1, x_2)] = \sum_{\mathbf{z}} \mathbb{E}[Y \mid \mathrm{do}(x_2), \mathbf{z}, x_1]P(\mathbf{z} \mid \mathrm{do}(x_1)), \tag{B.5}$$

which is an expectation of $Y$ over a product of probability measure

$$P(Y \mid \mathbf{Z}, \mathrm{do}(x_2), X_1)P(\mathbf{Z} \mid \mathrm{do}(x_1))\mathbb{1}_{x_1}(X_1),$$

which satisfies an additivity. Therefore, $\mathbb{E}[Y \mid \mathrm{do}(x_1, x_2)]$ is UCA-expressible. Such reduction can be done since the probability measure satisfies additivity w.r.t. all conditional distributions and the policy $\mathbb{1}_{x_1}(X_1)$. Specifically, set

$$\mathbf{C}_1 := \mathbf{Z}$$
$$\mathbf{R}_1 := \{X_1\}$$
$$\mathbf{S}_1 := \{X_1\} \cup \mathbf{Z}.$$

Also, set

$$P^1(\mathbf{C}_1) := P(\mathbf{Z} \mid \mathrm{do}(x_1))$$
$$P^2(Y \mid \mathbf{C}_1 \cup \mathbf{R}_1) := P(Y \mid X_1, \mathbf{Z}, \mathrm{do}(x_2))$$
$$\sigma^1_{\mathbf{R}_1} := \mathbb{1}_{x_1}(X_1).$$

The regression nuisances are the followings:

$$\mu_0^1(\mathbf{C}^{(1)} \cup \mathbf{R}^{(1)}) := \mu_0^1(X_1, \mathbf{Z}) := \mathbb{E}_P[Y \mid X_1, \mathbf{Z}, \mathrm{do}(x_2)]$$
$$\check{\mu}_0^1(\mathbf{C}^{(1)}) := \mathbb{E}_P[Y \mid x_1, \mathbf{Z}, \mathrm{do}(x_2)].$$

The ratio nuisances are the followings:

$$\pi_0^1(X, \mathbf{Z}) = \frac{\mathbb{1}_{x_1}(X_1) P(\mathbf{Z} \mid \mathrm{do}(x_1))}{P(X_1 \mid \mathbf{Z}, \mathrm{do}(x_2)) P(\mathbf{Z} \mid \mathrm{do}(x_2))},$$

which can be estimated through the density estimation approach using the probabilistic classification method described in (Díaz et al., 2023, Sec. 5.4).

The representation for DML-UCA is

$$\mathbb{E}_P[\pi_0^1(X, \mathbf{Z})\{Y - \mu_0^1(X, \mathbf{Z})\} \mid \mathrm{do}(x_2)] + \mathbb{E}_P[\check{\mu}_0^1(\mathbf{Z}) \mid \mathrm{do}(x_1)].$$

## C   More Results

### C.1   Formal definition of Partial influence function (PIF)

**Definition C.1** (**Partial influence function (PIF)** (Pires and Branco, 2002)). *Let $g(\mathtt{P}^1, \cdots, \mathtt{P}^K)$ denote a $K$-multi-distribution functional. For the $k$-th component, let $\mathtt{P}_t^k := \mathtt{P}^k + t(\mathtt{Q}^k - \mathtt{P}^k)$ for $t \in [0, 1]$, where $\mathtt{Q}^k$ is an arbitrary distribution absolutely continuous w.r.t. $\mathtt{P}^k$. The $k$-th* **partial influence function** *is a function $\phi^k(\mathbf{V}; \boldsymbol{\eta}^i(\mathtt{P}^k), g_0)$ such that $\mathbb{E}_{\mathtt{P}^k}[\phi^k(\mathbf{V}; \eta^k(\mathtt{P}^k), g_0)] = 0$, $\mathbb{V}_{\mathtt{P}^k}[\phi^k(\mathbf{V}; \eta^k(\mathtt{P}^k), g_0)] < \infty$, and $\frac{\partial}{\partial t} g(\mathtt{P}^1, \cdots, \mathtt{P}_t^k, \cdots, \mathtt{P}^K)\big|_{t=0} = \mathbb{E}_{\mathtt{Q}^k}[\phi^k(\mathbf{V}; \boldsymbol{\eta}^k(\mathtt{P}^k), g_0)].$*

### C.2   Density Ratio Estimation

Two available approaches for estimating the density ratio are the followings. The first approach is to apply the Bayes rule for rewriting the density ratio into more tractable form. For example, consider the problem of estimating $\pi_0^2$ for FD, which is given as

$$\pi_0^2 := \frac{P(Z \mid x, C)}{P(Z \mid X, C)}.$$

Suppose $Z, C$ are high-dimensional random vectors, and $X$ is a binary singleton variable. Then, $P(X \mid C)$ or $P(X \mid Z, C)$ are tractable to estimate compared to $P(Z \mid X, C)$, since estimating $P(X \mid \cdot)$ can be done using off-the-shelf probabilistic classification method. Here, $\pi_0^2$ can be written

as a tractable form as follows:

$$\pi_0^2 := \frac{P(Z \mid x, C)}{P(Z \mid X, C)}$$

$$= \frac{P(Z, X, C)}{P(X \mid C)P(C)} \frac{P(x \mid C)P(C)}{P(Z, x, C)}$$

$$= \frac{P(C)}{P(C)} \frac{P(Z, C)}{P(Z, C)} \frac{P(x \mid C)}{P(X \mid C)} \frac{P(X \mid Z, C)}{P(x \mid Z, C)}$$

$$= \frac{P(x \mid C)}{P(X \mid C)} \frac{P(X \mid Z, C)}{P(x \mid Z, C)}.$$

The second approach is to recast the density ratio into the classification problem (Díaz et al., 2023, Sec. 5.4). For example, consider the ratio nuisance appeared in Treatment-treatment interactions:

$$\pi_0^1(X, \mathbf{Z}) = \frac{\mathbb{1}_{x_1}(X_1)P(\mathbf{Z} \mid \mathrm{do}(x_1))}{P(X_1 \mid \mathbf{Z}, \mathrm{do}(x_2))P(\mathbf{Z} \mid \mathrm{do}(x_2))}.$$

Here, $\frac{P(\mathbf{Z} \mid \mathrm{do}(x_1))}{P(\mathbf{Z} \mid \mathrm{do}(x_2))}$ can be estimated as a following procedure. Let $\mathcal{D}_1 \sim P(\mathbf{Z} \mid \mathrm{do}(x_1)$ and $\mathcal{D}_2 \sim P(\mathbf{Z} \mid \mathrm{do}(x_2)$ denote samples. Let $\mathcal{D}_0 := \mathcal{D}_1 \cup \mathcal{D}_2$. Let $\lambda$ denote an indicator such that $\lambda = 0$ means samples are from $\mathcal{D}_1$ and $\lambda = 1$ means they are from $\mathcal{D}_2$. Without loss of generality, $|\mathcal{D}_1| = |\mathcal{D}_2|$. Then,

$$\frac{P(\mathbf{Z} \mid \mathrm{do}(x_1))}{P(\mathbf{Z} \mid \mathrm{do}(x_2))} = \frac{P(\mathbf{Z} \mid \lambda = 0)}{P(\mathbf{Z} \mid \lambda = 1)} = \frac{P(\lambda = 1)}{P(\lambda = 0)} \frac{P(\lambda = 0 \mid \mathbf{Z})P(\mathbf{Z})}{P(\lambda = 1 \mid \mathbf{Z})P(\mathbf{Z})} = \frac{P(\lambda = 0 \mid \mathbf{Z})}{P(\lambda = 1 \mid \mathbf{Z})}.$$

Then, instead of estimating the density ratio explicitly as $\frac{P(\mathbf{Z} \mid \mathrm{do}(x_1))}{P(\mathbf{Z} \mid \mathrm{do}(x_2))}$, we can estimate the equivalent estimand $\frac{P(\lambda = 0 \mid \mathbf{Z})}{P(\lambda = 1 \mid \mathbf{Z})}$ using any off-the-shelf probabilistic classification method.

### C.3   Analysis of non-UCA functionals

We consider two cases where a target estimand cannot be expressed through UCA:

1. **Case 1.** The target estimand is not in a form of the product (e.g., the target estimand is the quotient of sum-products of two conditional distributions ).

2. **Case 2.** For a target estimand that is represented as the expectation of $Y$ over the measure

$$\Psi'[\mathbf{P}; \boldsymbol{\sigma}] := P^{m+1}(Y \mid \mathbf{S}'_m) \prod_{i=1}^{m} P^i(\mathbf{C}_i \mid \mathbf{S}'_{i-1}) \sigma^i_{\mathbf{R}_i}(\mathbf{R}_i \mid \mathbf{S}'_i \setminus \mathbf{R}_i),$$

where $P^i(\mathbf{V}) = Q^i(\mathbf{V} \mid \mathbf{S}^b_{i-1} = \mathbf{s})$ for some distribution $Q^i$, $\exists i \in \{2, \cdots, m+1\}$ such that $\mathbf{S}'_{i-1} \neq (\mathbf{C}^{(i-1)} \cup \mathbf{R}^{(i-1)}) \setminus \mathbf{S}^b_{i-1}$.

In this section, we will provide example functionals that cannot be expressed through UCA.

### C.3.1   On Case 1

Consider Fig. 1c where the causal effect $P(y \mid \mathrm{do}(x))$ is identifiable and given as

$$P(y \mid \mathrm{do}(x)) = \frac{\sum_w P(y, x \mid r, w)P(w)}{\sum_w P(x \mid r, w)P(w)}. \tag{C.1}$$

Here, the functional for $\mathbb{E}[Y \mid \mathrm{do}(x)]$ is represented not as the expectation of a product of conditional distributions, but rather as a quotient of sums of conditional distributions. The napkin estimand is not UCA-expressible.

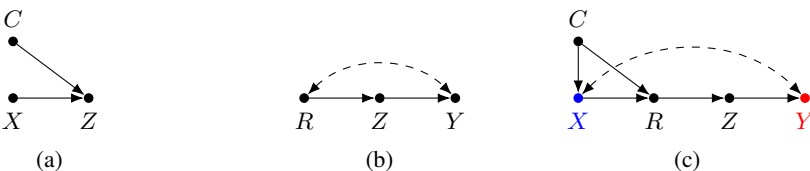

Figure C.3: **(a-c)** Example for Case 2 (Generalized identification under partial observability (Lee and Bareinboim, 2020, Fig. 1))

### C.3.2 On Case 2

Consider Figs. (C.3a-C.3c). A goal is to identify $P(y \mid \mathrm{do}(x))$ from Fig. C.3c from two input distributions: (1) an interventional distribution $P(c, z \mid \mathrm{do}(x))$ with Fig. C.3a, and (2) an observational distribution $P(r, z, y)$ with Fig. C.3b. This problem is entitled as the *generalized identification under partial observability* (Lee and Bareinboim, 2020).

Here, the causal effect is identifiable and given as (Lee and Bareinboim, 2020)

$$\mathbb{E}[Y \mid \mathrm{do}(x)] = \sum_{c,r,z} \mathbb{E}_P[Y \mid r, z]P(z \mid \mathrm{do}(x), c)P(r)P(c \mid \mathrm{do}(x)). \tag{C.2}$$

This functional is an expectation of the probability measure $Y$ over $P(Y \mid R, Z)P(Z \mid \mathrm{do}(x), C)P(R)P(C \mid \mathrm{do}(x))$. Based on this probability measure, apply the following setting:

- $\mathbf{C}_1 = \{C\}$, $\mathbf{C}_2 := \{R\}$ and $\mathbf{C}_3 := \{Z\}$.

- $\mathbf{R} = \emptyset$.

- $P^1(\mathbf{C}_1) := P(C \mid \mathrm{do}(x))$ with $\mathbf{S}_0^b = \emptyset$.

- $P^2(\mathbf{C}_2 \mid \mathbf{S}_1') = P(R)$ with $\mathbf{S}_1^b = \emptyset$ and $\mathbf{S}_1' := \emptyset$.

- $P^3(\mathbf{C}_3 \mid \mathbf{S}_2') = P(Z \mid \mathrm{do}(x), C)$ with $\mathbf{S}_2^b = \emptyset$ and $\mathbf{S}_2' := \{C\}$.

- $P^4(Y \mid \mathbf{S}_3') = P(Y \mid R, Z)$ with $\mathbf{S}_2^b = \emptyset$ and $\mathbf{S}_3' := \{R, Z\}$.

Here, $\mathbf{S}_1' = \emptyset \neq (\mathbf{C}^{(1)} \cup \mathbf{R}^{(1)}) \setminus \mathbf{S}_1^b = \{C\}$. Also, $\mathbf{S}_2' = \{C\} \neq (\mathbf{C}^{(2)} \cup \mathbf{R}^{(2)}) \setminus \mathbf{S}_2^b = \{C, R\}$. Finally, $\mathbf{S}_3' = \{R, Z\} \neq (\mathbf{C}^{(3)} \cup \mathbf{R}^{(3)}) \setminus \mathbf{S}_3^b = \{C, R, Z\}$. Therefore, Eq. (C.2) is not within UCA-class.

Now, we will witness that the target estimand cannot be correctly represented through the nested regression and empirical bifurcation. Applying the nested regression, we have

$$\mu_0^3(\mathbf{S}_3') = \mu_0^3(R, Z) := \mathbb{E}_P[Y \mid R, Z]$$
$$\check{\mu}_0^3(\check{\mathbf{S}}_3') = \mu_0^3(R, Z) = \mathbb{E}_P[Y \mid R, Z].$$

Then,

$$\mu_0^2(\mathbf{S}_2') = \mathbb{E}[\mu_0^3(R, Z) \mid C, \mathrm{do}(x)] = \sum_{r,z} \mathbb{E}_P[Y \mid r, z]P(r, z \mid c, \mathrm{do}(x)).$$

This doesn't correctly represent the target estimand in Eq. (C.2) because $P(r, z \mid c, \mathrm{do}(x))$ is not decomposed into $P(r)$ and $P(z \mid r, c, \mathrm{do}(x))$.

### C.4 Time Complexity

In this section, we provide a detailed analysis on the time complexity in Table 2. Here, $n$ is the sample size. When there are multiple sample sets (i.e., $K > 1$), we will use $n_{\max}$ to denote the size of the largest sample set. $K$ is the number of sample sets in Eq. (11). $m$ is the number of variables in a causal graph.

### C.4.1 BD/SBD

Here, we focus on the back-door adjustment, since the SBD can be analyzed similarly. The back-door (BD) adjustment estimand is given as

$$\sum_{\mathbf{z}} \mathbb{E}_P[Y \mid \mathbf{x}, \mathbf{z}] P(\mathbf{z}).$$

**Plug-in.** The plug-in estimator composed of two stage – learn the conditional probability table and and evaluate it for each samples. For a detailed description, we define some notations. Let $\mathcal{D} := \{\mathbf{V}_{(j)} : j = 1, \cdots, n\}$, where $\mathbf{V}_{(j)}$ denote the $j$'th sample. For any $\mathbf{W} \subseteq \mathbf{V}$, we will use $\mathcal{D}_{\mathbf{w}}$ to denote the sub-sample of $\mathcal{D}$ that $\mathbf{W}$ is fixed to $\mathbf{w}$; i.e., $\mathcal{D}_{\mathbf{w}} := \{\mathbf{V}_{(j)} \in \mathcal{D}$ such that $\mathbf{W}_{(j)} = \mathbf{w}\}$. We will use $\mathcal{I}(\mathcal{D}_{\mathbf{w}})$ to denote the index set for $\mathcal{D}_{\mathbf{w}}$. Finally, we will use $n_{\mathbf{w}} := |\mathcal{D}_{\mathbf{w}}|$.

For the BD adjustment, the plug-in estimator is

$$\sum_{\mathbf{z}} \hat{\mathbb{E}}[Y \mid \mathbf{x}, \mathbf{z}] \hat{P}(\mathbf{z}), \tag{C.3}$$

where

$$\hat{\mathbb{E}}[Y \mid \mathbf{x}, \mathbf{z}] := \frac{1}{n_{\mathbf{x}, \mathbf{z}}} \sum_{j \in \mathcal{I}(\mathcal{D}_{\mathbf{x}, \mathbf{z}})} Y_{(j)}, \tag{C.4}$$

$$\hat{P}(\mathbf{z}) = \frac{n_{\mathbf{z}}}{n}. \tag{C.5}$$

For the fixed $(\mathbf{x}, \mathbf{z})$, learning $\hat{\mathbb{E}}[Y \mid \mathbf{x}, \mathbf{z}]$ and $\hat{P}(\mathbf{z})$ take $O(n)$. Such learning needs to be done for all possible realizations $(\mathbf{x}, \mathbf{z})$, where the cardinality of the realization is $O(2^m)$. As a result, the computational complexity is $O(n2^m)$.

**IPW, OM, AIPW** We first consider the inverse probability-weighting (IPW) estimator (Rosenbaum and Rubin, 1983). The IPW estimator is

$$\frac{1}{n} \sum_{i=1}^{n} \frac{\mathbb{1}_{\mathbf{x}}(\mathbf{X}_{(i)})}{\hat{\pi}(\mathbf{X}_{(i)} \mid \mathbf{Z}_{(i)})} Y_{(i)}, \tag{C.6}$$

where $\hat{\pi}(\mathbf{X} \mid \mathbf{Z})$ is the evaluated function for $P(\mathbf{X} \mid \mathbf{Z})$. Learning the nuisance parameter $\hat{\pi}$ takes $T(n, m)$ and evaluating the IPW estimator takes $O(n)$. As a result, the time complexity for the IPW estimator is $O(n + T(n, m))$.

Next, we consider the outcome-model (OM) estimator (Robins, 1986). The OM estimator is

$$\frac{1}{n} \sum_{i=1}^{n} \hat{\mu}(\mathbf{X}_{(i)}, \mathbf{Z}_{(i)}), \tag{C.7}$$

where $\hat{\mu}(\mathbf{X}, \mathbf{Z})$ is the evaluated function for $\mathbb{E}_P[Y \mid \mathbf{X}, \mathbf{Z}]$. Learning the nuisance parameter $\hat{\pi}$ takes $T(n, m)$ and evaluating the OM estimator takes $O(n)$. As a result, the time complexity for the OM estimator is $O(n + T(n, m))$.

Finally, the AIPW estimator (Robins and Rotnitzky, 1995) is

$$\frac{1}{n} \sum_{i=1}^{n} \frac{\mathbb{1}_{\mathbf{x}}(\mathbf{X}_{(i)})}{\hat{\pi}(\mathbf{X}_{(i)} \mid \mathbf{Z}_{(i)})} Y_{(i)} + \frac{1}{n} \sum_{i=1}^{n} \hat{\mu}(\mathbf{X}_{(i)}, \mathbf{Z}_{(i)}) - \frac{1}{n} \sum_{i=1}^{n} \frac{\mathbb{1}_{\mathbf{x}}(\mathbf{X}_{(i)})}{\hat{\pi}(\mathbf{X}_{(i)} \mid \mathbf{Z}_{(i)})} \hat{\mu}(\mathbf{X}_{(i)}, \mathbf{Z}_{(i)}). \tag{C.8}$$

Learning the nuisance parameters $\hat{\pi}$ and $\hat{\mu}$ takes $T(n, m)$ and evaluating the OM/IPW estimator takes $O(n)$. As a result, the time complexity for the OM estimator is $O(n + T(n, m))$.

### C.4.2 Front-door adjustment (FD)

The front-door adjustment (Pearl, 2000) is

$$\sum_{\mathbf{z},\mathbf{c}} P(\mathbf{z} \mid \mathbf{x}, \mathbf{z}) \sum_{\mathbf{z}} \mathbb{E}_P[Y \mid \mathbf{z}, \mathbf{x}', \mathbf{c}] P(\mathbf{x}' \mid \mathbf{c}) P(\mathbf{c}).$$

The FD estimators of (Fulcher et al., 2019; Guo et al., 2023) is

$$\frac{1}{n}\sum_{i=1}^{n} \frac{\hat{\xi}(\mathbf{Z}_{(i)}, x, \mathbf{C}_{(i)})}{\hat{\xi}(\mathbf{Z}_{(i)}, \mathbf{X}_{(i)}, \mathbf{C}_{(i)})}\{Y_{(i)} - \hat{\mu}(\mathbf{C}_{(i)}, \mathbf{X}_{(i)}, \mathbf{Z}_{(i)})\}$$

$$+ \frac{1}{n}\sum_{i=1}^{n} \frac{\mathbb{1}_{\mathbf{x}}(\mathbf{X}_{(i)})}{\hat{\pi}(\mathbf{X}_{(i)}, \mathbf{C}_{(i)})}\left\{\sum_{\mathbf{x}} \hat{\mu}(\mathbf{C}_{(i)}, \mathbf{x}, \mathbf{Z}_{(i)})\hat{\pi}(\mathbf{x}, \mathbf{C}_{(i)}) - \sum_{\mathbf{x},\mathbf{z}} \hat{\mu}(\mathbf{C}_{(i)}, \mathbf{x}, \mathbf{z})\hat{\xi}(\mathbf{z}, \mathbf{X}_{(i)}, \mathbf{C}_{(i)})\hat{\pi}(\mathbf{x}, \mathbf{C}_{(i)})\right\}$$

$$+ \frac{1}{n}\sum_{i=1}^{n}\{\sum_{\mathbf{z}} \hat{\mu}(\mathbf{C}_{(i)}, \mathbf{X}_{(i)}, \mathbf{z})\hat{\xi}(\mathbf{z}, \mathbf{x}, \mathbf{C}_{(i)})\}, \tag{C.9}$$

where $\hat{\mu}(\mathbf{C}, \mathbf{X}, \mathbf{Z})$, $\hat{\xi}(\mathbf{Z}, \mathbf{X}, \mathbf{C})$ and $\hat{\pi}(\mathbf{X}, \mathbf{C})$ are the evaluated functions for $\mathbb{E}_P[Y \mid \mathbf{C}, \mathbf{X}, \mathbf{Z}]$, $P(\mathbf{Z} \mid \mathbf{X}, \mathbf{C})$ and $P(\mathbf{X} \mid \mathbf{C})$. Learning these nuisances takes $T(n, m)$ time. Equipped with $\hat{\mu}, \hat{\xi}, \hat{\pi}$, evaluating the FD estimator takes $O(n2^m)$, since the evaluation over $n$ samples is repeated for $O(2^m)$ realization of every $(\mathbf{x}, \mathbf{z})$. Therefore, the overall time complexity is $O(n2^m + T(n, m))$.

### C.4.3 Tian's adjustment

The Tian's adjustment (Tian and Pearl, 2002a) is

$$\sum_{\mathbf{v}\backslash xy} \sum_{x'} \mathbb{E}_{P'}[Y \mid \mathbf{v}^{(K)}] \prod_{i=1}^{K} P'(v_i \mid \mathbf{v}^{(i-1)}).$$

The estimator for Tian's adjstment proposed in (Bhattacharya et al., 2022) is $\frac{1}{n}\sum_{i=1}^{n} \varphi(\mathbf{V}_{(i)}; \hat{\eta})$, where $\varphi(\mathbf{V}; \eta_0)$ is given as

$$\sum_{V_i \in \mathbf{V}^{\geq k+1}\backslash \mathbf{S}_X} \left\{ \prod_{j=1}^{i-1} \frac{\mathbb{1}_x(X)}{P(V_j \mid \mathbf{V}^{(j-1)})} \sum_{x',\mathbf{v}^{\geq i+1}} y \prod_{V_j \in (\mathbf{V}^{\geq k} \cap \mathbf{S}_X) \cup \mathbf{V}^{\geq i+1}} P(v_j \mid \mathbf{v}^{(j-1)})|_{x=x' \text{ if } V_j \notin \mathbf{S}_X} \right\}$$

$$- \sum_{V_i \in \mathbf{V}^{\geq k+1}\backslash \mathbf{S}_X} \left\{ \prod_{j=1}^{i-1} \frac{\mathbb{1}_x(X)}{P(V_j \mid \mathbf{V}^{(j-1)})} \sum_{x',\mathbf{v}^{\geq i+1}} y \prod_{V_j \in (\mathbf{V}^{\geq k} \cap \mathbf{S}_X) \cup \mathbf{V}^{\geq i}} P(v_j \mid \mathbf{v}^{(j-1)})|_{x=x' \text{ if } V_j \notin \mathbf{S}_X} \right\}$$

$$+ \sum_{V_i \in V_i \in \mathbf{V}^{\geq k+1} \cap \mathbf{S}_X} \frac{\prod_{V_j \in \mathbf{V}^{(i-1)}} P(V_j \mid \mathbf{V}^{(j-1)})|_{X=x}}{\prod_{V_j \in \mathbf{V}^{(i-1)}} P(V_j \mid \mathbf{V}^{(j-1)})} \left\{ \sum_{\mathbf{v}^{\geq i+1}} y \prod_{V_j \in \mathbf{V}^{\geq i+1}} P(v_j \mid \mathbf{v}^{(j-1)})|_{x=x' \text{ if } V_j \notin \mathbf{S}_X} \right\}$$

$$- \sum_{V_i \in V_i \in \mathbf{V}^{\geq k+1} \cap \mathbf{S}_X} \frac{\prod_{V_j \in \mathbf{V}^{(i-1)}} P(V_j \mid \mathbf{V}^{(j-1)})|_{X=x}}{\prod_{V_j \in \mathbf{V}^{(i-1)}} P(V_j \mid \mathbf{V}^{(j-1)})} \left\{ \sum_{\mathbf{v}^{\geq i}} y \prod_{V_j \in \mathbf{V}^{\geq i+1}} P(v_j \mid \mathbf{v}^{(j-1)})|_{x=x' \text{ if } V_j \notin \mathbf{S}_X} \right\}$$

$$+ \sum_{\mathbf{v}^{\geq k+1}} y \prod_{V_j \in \mathbf{V}^{\geq k+1}\backslash \mathbf{S}_X} P(v_j \mid \mathbf{v}^{(j-1)})|_{x'=x} \prod_{V_r \in \mathbf{V}^{\geq k+1} \cap \mathbf{S}_X} P(v_r \mid \mathbf{v}^{(r-1)}).$$

Learning these nuisances takes $T(n, m)$ time. Equipped with nuisances corresponding to $\hat{P}(v_i \mid \mathbf{v}^{(i-1)})$, evaluating the estimator takes $O(n2^m)$, since the evaluation over $n$ samples is repeated for $O(2^m)$ realization of every $\mathbf{v}$. Therefore, the overall time complexity is $O(n2^m + T(n, m))$.

### C.4.4 DML-UCA (BD, FD, and Tian's).

For BD, FD, and Tian's, the time complexity can be derived by specializing Thm. 2 with $K = 1$ and $n_{\max} = n$, and $T(m, n) \coloneqq K \times L \times (T_{\boldsymbol{\mu}} + T_{\boldsymbol{\pi}})$. Then, the complexity in Thm. 2 reduces to $O(n + T(m, n))$.

### C.4.5 DML-UCA (general).

Set $T(m, n_{\max}, K) \coloneqq L \times K \times (T_{\boldsymbol{\mu}} + T_{\boldsymbol{\pi}})$. Then, the complexity in Thm. 2 reduces to $O(Kn_{\max} + T(m, n_{\max}, K))$.

### C.4.6 DML-ID (obsID).

The DML-ID estimator of (Jung et al., 2021a) writes the identification functional of causal effect as am arithmetic function of multiple sequential back-door adjustments, where the arithmetic function is an arbitrary combination of marginalization, product, and division. Let $f(\{A^k : k = 1, \cdots, K\})$ denote the DML-ID, where each $A^k$ is the sequential back-door adjustment, and $f$ denotes the arithmetic function.

To evaluate the DML-ID functional, the first step is to learn all nuisances composing each $A^k$. This takes $O(T(m, n))$ time. The second step is to evaluate $f(\{A^k : k = 1, \cdots, K\})$. Whenever $f$ contains a marginalization over some random vector, the time complexity for evaluating it is $O(n2^m)$. In DML-ID, such marginalization can happen $O(2^m)$ times in worst case. Therefore, evaluating $f(\{A^k : k = 1, \cdots, K\})$ can take $O(n2^m \times 2^m) = O(n2^{2m})$. As a result, the total time complexity is $O(n2^{2m} + T(n, m))$.

### C.4.7 DML-gID (gID).

The DML-gID estimator of (Jung et al., 2023a) writes the identification functional of causal effect as am arithmetic function of multiple generalized sequential back-door adjustments called g-mSBD (Jung et al., 2023a), where the arithmetic function is an arbitrary combination of marginalization, product, and division. Let $f(\{A^j : j = 1, \cdots, J\})$ denote the DML-ID, where each $A^j$ is the g-mSBD, and $f$ denotes the arithmetic function.

To evaluate the DML-gID functional, the first step is to learn all nuisances composing each $A^j$. This takes $O(T(m, n_{\max}, K))$ time. The second step is to evaluate $f(\{A^j : j = 1, \cdots, J\})$. Whenever $f$ contains a marginalization over some random vector, the time complexity for evaluating it is $O(n_{\max}2^m)$. In DML-gID, such marginalization can happen $O(2^m \times K)$ times in worst case. Therefore, evaluating $f(\{A^j : j = 1, \cdots, J\})$ can take $O(n_{\max}2^m \times 2^m \times K) = O(Kn_{\max}2^{2m})$. As a result, the total time complexity is $O(Kn_{\max}2^{2m} + T(m, n_{\max}, K))$.

## D  Proofs

### D.1  Proof of Proposition 1

The algorithm provides a product of probabilities in a form of

$$\Psi[\mathbf{P}] \coloneqq \prod_{V_i \in \mathbf{S}_X} P(V_i \mid \mathbf{V}^{(i)}) \prod_{V_j \notin \mathbf{V} \setminus \mathbf{S}_X} P(V_j \mid \mathbf{V}^{(j)} \setminus X, x).$$

Then, by taking an expectation over $Y$, it gives

$$\sum_{\mathbf{v} \setminus xy} \sum_{x'} \mathbb{E}_{P'}[Y \mid \mathbf{v}^{(K)}] \prod_{i=1}^{K} P'(v_i \mid \mathbf{v}^{(i-1)}).$$

$\blacksquare$

## D.2 Proof for Proposition 2

First, define

$$\texttt{set}_1 := ((\mathbf{S}_{i+1} \cup \mathbf{B}_{i+1}) \setminus \mathbf{R}_{i+1} \setminus \mathbf{B}_i)$$
$$= (((\mathbf{C}^{(i+1)} \cup \mathbf{R}^{(i)} \cup \mathbf{B}_{i+1}) \setminus \mathbf{S}_{i+1}^b) \setminus \mathbf{B}_i) \cup \mathbf{B}_i',$$

and

$$\texttt{set}_2 := \mathbf{S}_i \cup \mathbf{B}_i' = (\mathbf{C}^{(i)} \cup \mathbf{R}^{(i)} \setminus \mathbf{S}_i^b) \cup \mathbf{B}_i'.$$

Then, we claim that $\texttt{set}_1 \setminus \texttt{set}_2 = \mathbf{C}_{i+1}$. This holds when $\mathbf{B}_{i+1} \setminus \mathbf{B}_i \setminus \mathbf{S}_i = \emptyset$. To prove this with contradiction, suppose $\mathbf{B}_{i+1} \setminus \mathbf{S}_i \neq \emptyset$. This holds when $\mathbf{B}_{i+1} \subseteq \mathbf{S}_i^b$. Recall that $\mathbf{B}_{i+1}$ is a subset of fixed variables in $\mathbf{S}_{i+1}^b$ in $P^{i+2}$. Then, $\mathbf{B}_{i+1} \subseteq \mathbf{S}_i^b$ means that this variable will be fixed in $P^{i+1}$. However, for this variable to be bifurcated in some $\mathbf{C}_j$, this variable should be within $\mathbf{B}_i$. However, this is a contradiction of the definition of $\mathbf{B}_{i+1}$ and $\mathbf{B}_i$. Therefore, $\mathbf{B}_{i+1} \setminus \mathbf{B}_i \setminus \mathbf{S}_i = \emptyset$ and $\texttt{set}_1 \setminus \texttt{set}_2 = \mathbf{C}_{i+1}$.

Then,

$$\mu_0^i(\mathbf{S}_i, \mathbf{B}_i') = \mathbb{E}_{P^{i+1}}[\check{\mu}_0^{i+1}(\check{\mathbf{S}}_{i+1}) \mid \mathbf{S}_i, \mathbf{B}_i']$$
$$= \mathbb{E}_{P^{i+1}}[\check{\mu}_0^{i+1}(((\mathbf{S}_{i+1} \cup \mathbf{B}_{i+1}) \setminus \mathbf{R}_{i+1} \setminus \mathbf{B}_i) \cup \mathbf{B}_i') \mid \mathbf{S}_i, \mathbf{B}_i']$$
$$= \sum_{\mathbf{c}_{i+1}, \mathbf{r}_{i+1}} P^{i+1}(\mathbf{c}_{i+1} \mid \mathbf{S}_i, \mathbf{B}_i') \sigma_{\mathbf{R}_{i+1}}^{i+1}(\mathbf{r}_{i+1}) \mu_0^{i+1}(\mathbf{c}_{i+1}, \mathbf{r}_{i+1}, \mathbf{S}_i, \mathbf{B}_i')$$
$$= \sum_{\mathbf{c}_{i+1}, \mathbf{r}_{i+1}} P^{i+1}(\mathbf{c}_{i+1} \mid \mathbf{S}_i) \sigma_{\mathbf{R}_{i+1}}^{i+1}(\mathbf{r}_{i+1}) \mu_0^{i+1}(\mathbf{c}_{i+1}, \mathbf{r}_{i+1}, \mathbf{S}_i, \mathbf{B}_i').$$

By recursion,

$$\mathbb{E}_{P^1}[\check{\mu}_0^1(\check{\mathbf{S}}_1)] = \sum_{\mathbf{c}^{(m)}, \mathbf{r}^{(m)}} \prod_{j=1}^{m} P^j(\mathbf{c}_j \mid \mathbf{s}_{j-1}) \sigma_{\mathbf{R}_j}^j(\mathbf{r}_j \mid \mathbf{s}_j \setminus \mathbf{r}_j) \mathbb{E}_{P^{m+1}}[Y \mid \mathbf{s}_m].$$

■

## D.3 Proof for Proposition 3

By definition of $\pi_0^m$. ■

## D.4 Proof for Theorem 1

If conditions in Theorem 1 met, the estimand reduces to the UCA by definition. ■

## D.5 Proof for Theorem 2

1. The sample-splitting takes $O((m+1)n_{\max})$.
2. For the fixed $\ell$, learning $\hat{\mu}_\ell^i$ for $i = m, \cdots, 1$ takes $O(T_{\boldsymbol{\mu}} \times m)$. Therefore, learning all regression-nuisances takes $O(T_{\boldsymbol{\mu}} \times m \times L)$.
3. For the fixed $\ell$, learning $\hat{\pi}_\ell^i$ for $i = 1, \cdots, m$ takes $O(T_{\boldsymbol{\pi}} \times m)$. Therefore, learning all ratio-nuisances takes $O(T_{\boldsymbol{\pi}} \times m \times L)$.
4. Evaluating the DML estimator in Eq. (8) takes $O((m+1)n_{\max})$.

In total, the time complexity is

$$O((m+1)n_{\max}) + O(T_{\boldsymbol{\mu}} \times m \times L) + O(T_{\boldsymbol{\pi}} \times m \times L) + O((m+1)n_{\max})$$
$$= O(m \times \{n_{\max} + L \times (T_{\boldsymbol{\mu}} + T_{\boldsymbol{\pi}})\})$$

■

## D.6  Proof for Theorem 3

Define $\Psi^i := \prod_{k=1}^{i} P^k(\mathbf{C}_k \mid \mathbf{S}_{k-1})\sigma^k(\mathbf{R}_k \mid \mathbf{S}_k \setminus \mathbf{R}_k)$. Define

$$\Psi_t^i := P_t^i(\mathbf{C}_i \mid \mathbf{S}_{i-1})\sigma^i(\mathbf{R}_i \mid \mathbf{S}_i \setminus \mathbf{R}_i)\prod_{k=1}^{i-1} P^k(\mathbf{C}_k \mid \mathbf{S}_{k-1})\sigma^k(\mathbf{R}_k \mid \mathbf{S}_k \setminus \mathbf{R}_k).$$

Define

$$\mu_t^i(\mathbf{S}_i') := \mathbb{E}_{P_t^{i+1}}[\check{\mu}^{i+1} \mid \mathbf{S}_i'].$$

For any $P^i$, we choose the following parametric submodel:

$$P_t^i := P^i + t(Q^i - P^i).$$

For any $i = m, \cdots, 1$,

$$\psi_0 := \mathbb{E}_{\Psi_i}[\mu_0^i(\mathbf{S}_i')].$$

Fix $i \in \{1, \cdots, m\}$. Then, consider the differention with respect to $P_t^{i+1}$.

$$
\begin{aligned}
\frac{\partial}{\partial t}\mathbb{E}_{\Psi_0^i}[\mu_t^i(\mathbf{S}_i')] &= \frac{\partial}{\partial t}\mathbb{E}_{P^{i+1}}[\mu_t^i(\mathbf{S}_i')\pi_0^i] \\
&= \frac{\partial}{\partial t}\mathbb{E}_{P_t^{i+1}}[\mu_t^i(\mathbf{S}_i')\pi_0^i] - \frac{\partial}{\partial t}\mathbb{E}_{P_t^{i+1}}[\mu_0^i(\mathbf{S}_i')\pi_0^i] \\
&= \frac{\partial}{\partial t}\mathbb{E}_{P^{i+1}}[\pi_0^i\check{\mu}^{i+1}(\check{\mathbf{S}}_{i+1})] - \frac{\partial}{\partial t}\mathbb{E}_{P_t^{i+1}}[\pi_0^i\mu_0^i(\mathbf{S}_i')] \\
&= \frac{\partial}{\partial t}\mathbb{E}_{P_t^{i+1}}[\pi_0^i\{\check{\mu}^{i+1}(\check{\mathbf{S}}_{i+1}) - \mu_0^i(\mathbf{S}_i')\}] \\
&= \mathbb{E}_{Q^{i+1}}[\pi_0^i\{\check{\mu}^{i+1}(\check{\mathbf{S}}_{i+1}) - \mu_0^i(\mathbf{S}_i')\}].
\end{aligned}
$$

Also, consider the differentiation with respect to $P^1$:

$$\frac{\partial}{\partial t}\mathbb{E}_{P_t^1}[\check{\mu}_0^1(\check{\mathbf{S}}_1)] = \mathbb{E}_{Q^1}[\check{\mu}_0^1(\check{\mathbf{S}}_1) - \psi_0].$$

Then, define $\varphi^i$ as the differentiation with respect to $P^i$ as

$$\varphi^i(\check{\mathbf{S}}_i; \eta_0^i, \psi_0) := \begin{cases} \pi_0^{i-1}\{\check{\mu}_0^i - \mu_0^{i-1}\} & \text{if } i > 1 \\ \check{\mu}_0^1 - \psi_0 & \text{if } i = 1. \end{cases}$$

Then,

$$
\begin{aligned}
\frac{\partial}{\partial t}\Psi(\mathbf{P}^1, \cdots, \mathbf{P}_t^k, \cdots, \mathbf{P}^K)\bigg|_{t=0} &= \sum_{i \in \mathcal{I}_k} \frac{\partial P_t^i}{\partial t}\frac{\partial}{\partial P_t^i}\Psi(P^1, \cdots, P_t^i, \cdots, P^{m+1}; \boldsymbol{\sigma})\bigg|_{t=0} \\
&= \sum_{i \in \mathcal{I}_k} \mathbb{E}_{Q^i}[\varphi^i(\check{\mathbf{S}}_i; \eta_0^i, \psi_0)],
\end{aligned}
$$

which completes the proof. ■

## D.7  Proof for Theorem 4

**Structure of the proof.**  Theorem 4 will be proven based on Lemma D.2, Lemma D.3, and Lemma D.4. Specifically, we proceed the proof as follows:

1. We will prove Lemma D.2, Lemma D.3, and Lemma D.4.
2. Berry-Essen's inequality (Berry, 1941) will be stated as a preliminary in Prop. D.1.
3. Theorem 4 will be proven based on the main lemmas and Berry-Essen's inequality.

### D.7.1 Helper lemmas

We first state and prove helper lemmas.

**Lemma D.1.**

$$\psi_0 = \sum_{i=1}^m \mathbb{E}_{P^{i+1}}[\pi_0^i \{\check{\mu}_0^{i+1} - \mu_0^i\}] + \mathbb{E}_{P^1}[\check{\mu}_0^1]. \tag{D.1}$$

**Proof of Lemma D.1.** By the total expectation law, it suffices to show that

$$\Psi(\mathbf{P}; \boldsymbol{\sigma}) = \mathbb{E}_{P^1}[\check{\mu}_0^1(\check{\mathbf{S}}_1)].$$

This holds from Prop. 2.

$\square$

**Lemma D.2** (**Decomposition**). *Define the following*

$$\Phi(\hat{\boldsymbol{\mu}}, \hat{\boldsymbol{\pi}}) := \sum_{i=1}^m \mathbb{E}_{P^{i+1}}[\hat{\pi}^i \{\check{\mu}^{i+1} - \hat{\mu}^i\}] + \mathbb{E}_{P^1}[\check{\mu}^1] \tag{D.2}$$

$$\Phi(\boldsymbol{\mu_0}, \boldsymbol{\pi_0}) := \sum_{i=1}^m \mathbb{E}_{P^{i+1}}[\pi_0^i \{\check{\mu}_0^{i+1} - \mu_0^i\}] + \mathbb{E}_{P^1}[\check{\mu}_0^1]. \tag{D.3}$$

*The following decomposition holds:*

$$\Phi(\hat{\boldsymbol{\mu}}, \hat{\boldsymbol{\pi}}) - \Phi(\boldsymbol{\mu_0}, \boldsymbol{\pi_0}) = \sum_{r=1}^m \mathbb{E}_{P^{r+1}}[\hat{\omega}^{(r-1)} \{\mu_0^r - \hat{\mu}^r\} \{\hat{\pi}^r - \pi_0^r\}]. \tag{D.4}$$

**Proof of Lemma D.2.** First,

$$\Phi(\hat{\boldsymbol{\mu}}, \hat{\boldsymbol{\pi}}) - \psi_0 = \Phi(\hat{\boldsymbol{\mu}}, \hat{\boldsymbol{\pi}}) - \Phi(\boldsymbol{\mu_0}, \boldsymbol{\pi_0}).$$

Also,

$$\mathbb{E}_{P^{m+1}}[\hat{\pi}^m \{\mu_0^m - \hat{\mu}^m\}] + \mathbb{E}_{P^{m+1}}[\pi_0^m \hat{\mu}^m] - \underbrace{\mathbb{E}_{P^{m+1}}[\pi_0^m \mu_0^m]}_{:=\psi_0}$$

$$= \mathbb{E}_{P^{m+1}}[\{\pi_0^m - \hat{\pi}^m\} \{\hat{\mu}^m - \mu_0^m\}].$$

For $i = m - 1, \cdots, 1$, define

$$\mu_0^i[\check{\mu}^{i+1}] := \mathbb{E}_{P^{i+1}}[\check{\mu}^{i+1}(\check{\mathbf{S}}_{i+1}) \mid \mathbf{S}_i, \mathbf{B}_i'].$$

Then,

$$\mathbb{E}_{P^{i+1}}[\hat{\pi}^i \{\mu_0^i[\check{\mu}^{i+1}] - \hat{\mu}^i\}] + \mathbb{E}_{P^{i+1}}[\pi_0^i \hat{\mu}^i] - \mathbb{E}_{P^{i+1}}[\pi_0^i \mu_0^i[\check{\mu}^{i+1}]]$$

$$= \mathbb{E}_{P^{i+1}}[\{\pi_0^i - \hat{\pi}^i\} \{\hat{\mu}^i - \mu_0^i[\check{\mu}^{i+1}]\}].$$

Also, for any $\mu^{i+1}$ and corresponding $\check{\mu}^{i+1}$, and for all $i = m - 1, \cdots, 1$, by the definition of the $\pi_0^i$ nuisance,

$$\mathbb{E}_{P^{i+2}}[\pi_0^{i+1} \mu^{i+1}] = \mathbb{E}_{P^{i+1}}[\pi_0^i \mu_0^i[\check{\mu}^{i+1}]].$$

Then,

$$\mathbb{E}_{P^{m+1}}[\hat{\pi}^m\{\mu_0^m - \hat{\mu}^m\}] + \mathbb{E}_{P^{m+1}}[\pi_0^m\hat{\mu}^m] - \mathbb{E}_{P^{m+1}}[\pi_0^m\mu_0^m]$$

$$+ \sum_{i=1}^{m-1}\mathbb{E}_{P^{i+1}}[\hat{\pi}^i\{\mu_0^i[\check{\mu}^{i+1}] - \hat{\mu}^i\}] + \mathbb{E}_{P^{i+1}}[\pi_0^i\hat{\mu}^i] - \mathbb{E}_{P^{i+1}}[\pi_0^i\mu_0^i[\check{\mu}^{i+1}]]$$

$$= \sum_{i=1}^{m}\mathbb{E}_{P^{i+1}}[\hat{\pi}^i\{\mu_0^i[\check{\mu}^{i+1}] - \hat{\mu}^i\}] + \mathbb{E}_{P^2}[\pi_0^1\hat{\mu}^1] - \psi_0$$

$$= \sum_{i=1}^{m}\mathbb{E}_{P^{i+1}}[\{\pi_0^i - \hat{\pi}^i\}\{\hat{\mu}^i - \mu_0^i[\check{\mu}^{i+1}]\}].$$

Note that $\mathbb{E}_{P^2}[\pi_0^1\hat{\mu}^1] = \mathbb{E}_{P^1}[\check{\mu}^1]$, since

$$\mathbb{E}_{P^1}[\check{\mu}^1(\mathbf{S}_1 \cup \mathbf{B}_1)]$$

$$= \mathbb{E}_{P^1}[\sigma_{\mathbf{R}_1}^1(\mathbf{R}_1 \mid \mathbf{S}_1 \setminus \mathbf{R}_1)\mu^1((\mathbf{S}_1 \cup \mathbf{B}_1) \setminus \mathbf{R}_1)]$$

$$= \mathbb{E}_{P^2}\left[\frac{P^1((\mathbf{S}_1 \cup \mathbf{B}_1) \setminus \mathbf{R}_1)}{P^2(\mathbf{S}_1 \cup \mathbf{B}_1)}\sigma_{\mathbf{R}_1}^1(\mathbf{R}_1 \mid \mathbf{S}_1 \setminus \mathbf{R}_1)\mu^1((\mathbf{S}_1 \cup \mathbf{B}_1) \setminus \mathbf{R}_1)\right]$$

$$= \mathbb{E}_{P^2}\left[\frac{P^1(\mathbf{C}_1)}{P^2(\mathbf{S}_1 \cup \mathbf{B}_1)}\sigma_{\mathbf{R}_1}^1(\mathbf{R}_1 \mid \mathbf{S}_1 \setminus \mathbf{R}_1)\mu^1((\mathbf{S}_1 \cup \mathbf{B}_1) \setminus \mathbf{R}_1)\right]$$

$$= \mathbb{E}_{P^2}[\pi_0^1\mu_0^1].$$

Therefore,

$$\Phi(\hat{\boldsymbol{\mu}}, \hat{\boldsymbol{\pi}}) - \Phi(\boldsymbol{\mu_0}, \boldsymbol{\pi_0})$$

$$= \sum_{i=1}^{m}\mathbb{E}_{P^{i+1}}[\{\pi_0^i - \hat{\pi}^i\}\{\hat{\mu}^i - \mu_0^i[\check{\mu}^{i+1}]\}].$$

$\square$

---

**Lemma D.3** (**Stochastic Equicontinuity**). *Let $\mathcal{D} \overset{iid}{\sim} P$. Let $\mathcal{D} = \mathcal{D}_0 \uplus \mathcal{D}_1$, where $n := |\mathcal{D}_0|$. Let $\hat{f}$ be a function estimated from $\mathcal{D}_1$. Then, in probability greater than $1 - \epsilon$ for any $\epsilon \in (0, 1)$,*

$$\mathbb{E}_{\mathcal{D}_0 - P}\left[\left|\hat{f} - f\right|\right] \overset{w.p\,1-\epsilon}{<} \frac{\|\hat{f} - f\|_P}{\sqrt{n\epsilon}}, \tag{D.5}$$

*which implies that*

$$\mathbb{E}_{\mathcal{D}_0 - P}[|\hat{f} - f|] = O_P\left(\frac{\|\hat{f} - f\|_P}{\sqrt{n}}\right).$$

---

***Proof of Lemma D.3.*** This proof is from (Kennedy et al., 2020, Lemma 2). Since $\hat{f}$ is a function of $\mathcal{D}_1$, we will denote $\hat{f}_{\mathcal{D}_1}$. Define a following random variable of interest:

$$X := \mathbb{E}_{\mathcal{D}_0 - P}[\hat{f}_{\mathcal{D}_1} - f].$$

Then, the conditional expectation of $X$ given $\mathcal{D}_1$ is zero, since

$$\mathbb{E}_P\left[\frac{1}{n}\sum_{i=1}^{n}\hat{f}_{\mathcal{D}_1}(\mathbf{V}_i) \,\middle|\, \mathcal{D}_1\right] = \frac{1}{n}\sum_{i=1}^{n}\mathbb{E}_P[\hat{f}_{\mathcal{D}_1}(\mathbf{V}_i) \mid \mathcal{D}_1] = \frac{1}{n}\sum_{i=1}^{n}\mathbb{E}_P[\hat{f}_{\mathcal{D}_1}(\mathbf{V}) \mid \mathcal{D}_1] = \mathbb{E}_P[\hat{f}_{\mathcal{D}_1}(\mathbf{V}) \mid \mathcal{D}_1],$$

where the third equality holds by the independence of $\mathcal{D}_0$ and $\mathcal{D}_1$. Therefore,

$$
\begin{aligned}
\mathbb{E}_P[X \mid \mathcal{D}_1] &= \mathbb{E}_P[\mathbb{E}_{\mathcal{D}_0 - P}[\hat{f}_{\mathcal{D}_1} - f] \mid \mathcal{D}_1] \\
&= \mathbb{E}_P[\mathbb{E}_{\mathcal{D}_0}[\hat{f}_{\mathcal{D}_1} - f] \mid \mathcal{D}_1] - \mathbb{E}_P[\mathbb{E}_P[\hat{f}_{\mathcal{D}_1} - f] \mid \mathcal{D}_1] \\
&= \mathbb{E}_P[\mathbb{E}_P[\hat{f}_{\mathcal{D}_1} - f] \mid \mathcal{D}_1] - \mathbb{E}_P[\mathbb{E}_P[\hat{f}_{\mathcal{D}_1} - f] \mid \mathcal{D}_1] = 0.
\end{aligned}
$$

Also,

$$
\begin{aligned}
\mathbb{V}_P[X \mid \mathcal{D}_1] &= \mathbb{V}_P[\mathbb{E}_{\mathcal{D}_0 - P}[\hat{f}_{\mathcal{D}_1} - f] \mid \mathcal{D}_1] \\
&= \mathbb{V}_P[\mathbb{E}_{\mathcal{D}_0}[\hat{f}_{\mathcal{D}_1} - f] \mid \mathcal{D}_1] \\
&= \frac{1}{n}\mathbb{V}_P[\hat{f}_{\mathcal{D}_1} - f \mid \mathcal{D}_1] \\
&\leq \frac{1}{n}\|\hat{f}_{\mathcal{D}_1} - f\|_P^2.
\end{aligned}
$$

By applying the (conditional-) Chevyshev's inequality,

$$
P(|X - \mathbb{E}_P[X \mid \mathcal{D}_1]| \geq t \mid \mathcal{D}_1) \leq \frac{1}{t^2}\mathbb{V}_P[X \mid \mathcal{D}_1] \leq \frac{1}{nt^2}\|\hat{f}_{\mathcal{D}_1} - f\|_P^2.
$$

Then,

$$
\begin{aligned}
P(|X| \geq t) &= P(|X - \mathbb{E}_P[X \mid \mathcal{D}_1]| \geq t) \\
&= \mathbb{E}_{P(\mathcal{D}_1)}[P(|X - \mathbb{E}_P[X \mid \mathcal{D}_1]| \geq t \mid \mathcal{D}_1)] \\
&\leq \frac{1}{nt^2}\|\hat{f}_{\mathcal{D}_1} - f\|_P^2.
\end{aligned}
$$

In other words, $X < t$ in probability greater than $1 - \frac{1}{nt^2}\|\hat{f}_{\mathcal{D}_1} - f\|_P^2$. If $t = \frac{\|\hat{f}_{\mathcal{D}_1} - f\|_P}{\sqrt{n\epsilon}}$, then $X < \frac{\|\hat{f}_{\mathcal{D}_1} - f\|_P}{\sqrt{n\epsilon}}$ in the probability greater than $1 - \epsilon$ for any $\epsilon \in (0, 1)$. $\qquad\square$

---

**Lemma D.4 (Combining concentration inequalities).** *Suppose $P(A_k > t) \leq b_k/t^2$ for $k = 1, \cdots, K$. Then,*

$$
P\left(\sum_{k=1}^K A_k \leq tK\right) \geq 1 - \frac{1}{t^2}\sum_{k=1}^K b_k.
$$

---

*Proof.* The event $\sum_{k=1}^K A_k \leq tK$ includes the case where $A_k < t$ for $k = 1, \cdots, K$. Therefore,

$$
\begin{aligned}
P\left(\sum_{k=1}^K A_k \leq tK\right) &\geq P\left(A_1 \leq t \text{ and } \cdots \text{ and } A_K \leq t\right) \\
&= 1 - P\left(A_1 > t \text{ or } \cdots \text{ or } A_K > t\right) \\
&\geq 1 - \sum_{k=1}^K P\left(A_k > t\right) \\
&\geq 1 - \sum_{k=1}^K \frac{b_k}{t^2}.
\end{aligned}
$$

$\square$

### D.7.2 Preliminary Results

**Proposition D.1** (**Berry–Esseen's inequality** (Berry, 1941; Esseen, 1942; Shevtsova, 2014)). *Suppose $\mathcal{D} = \{X_1, \cdots, X_n\}$ are independent and identically distributed random variables with $\mathbb{E}_P[X_i] = 0$, $\mathbb{E}_P[X_i^2] = \sigma^2$ and $\mathbb{E}_P[|X_i|^3] = \kappa^3$. Then, for all $x$ and $n$,*

$$\left| P\left( \frac{\sqrt{n}}{\sigma_0} \mathbb{E}_{\mathcal{D}}[X] < x \right) - \Phi(x) \right| \leq \frac{0.4748\kappa^3}{\sigma^3 \sqrt{n}}.$$

### D.7.3 Proof of Theorem 4 - (1)

By Lemma D.2, we decompose the error as follow:

$$\hat{\psi} - \psi_0 = \sum_{k=1}^{K} \mathbb{E}_{\mathcal{D}^k - \mathrm{P}^k}[\phi_0^k] \tag{D.6}$$

$$+ \frac{1}{L} \sum_{\ell=1}^{L} \sum_{k=1}^{K} \mathbb{E}_{\mathcal{D}_\ell^k - \mathrm{P}^k}[\hat{\phi}_\ell^k - \phi_0^k] \tag{D.7}$$

$$+ \frac{1}{L} \sum_{\ell=1}^{L} \sum_{i=1}^{m} \mathbb{E}_{P^{i+1}}[\{\mu_0^i - \hat{\mu}_\ell^i\}\{\hat{\pi}_\ell^i - \pi_0^i\}]. \tag{D.8}$$

Define

$$R_1^k := \mathbb{E}_{\mathcal{D}^k - \mathrm{P}^k}[\phi_0^k] + \frac{1}{L} \sum_{\ell=1}^{L} \mathbb{E}_{\mathcal{D}_\ell^k - \mathrm{P}^k}[\hat{\phi}_\ell^k - \phi_0^k].$$

Then it completes the proof.

### D.7.4 Proof of Theorem 4 - (2)

We first study the term $\mathbb{E}_{\mathcal{D}^k - \mathrm{P}^k}[\phi_0^k]$. By Chebyshev's inequality,

$$P\left( \left| \mathbb{E}_{\mathcal{D}^k - \mathrm{P}^k}[\phi_0^k] \right| > t \frac{\rho_{k,0}}{\sqrt{|\mathcal{D}^k|}} \right) < \frac{1}{t^2}.$$

Equivalently,

$$P\left( \left| \mathbb{E}_{\mathcal{D}^k - \mathrm{P}^k}[\phi_0^k] \right| > t \right) < \frac{1}{t^2} \frac{\rho_{k,0}^2}{|\mathcal{D}^k|}.$$

By Lemma D.4,

$$P\left( \sum_{k=1}^{K} \left| \mathbb{E}_{\mathcal{D}^k - \mathrm{P}^k}[\phi_0^k] \right| \leq t_1 K \right) > 1 - \frac{1}{t_1^2} \sum_{k=1}^{K} \frac{\rho_{k,0}^2}{|\mathcal{D}^k|}.$$

By Lemma D.3,

$$\mathrm{P}^k\left( \left| \mathbb{E}_{\mathcal{D}_\ell^k - \mathrm{P}^k}[\hat{\phi}_\ell^k - \phi_0^k] \right| > t_2 \right) \leq \frac{1}{t_2^2} \frac{\|\hat{\phi}_\ell^k - \phi_0^k\|_{\mathrm{P}^k}^2}{|\mathcal{D}_\ell^k|}. \tag{D.9}$$

By Lemma D.4,

$$P\left( \frac{1}{L} \sum_{\ell=1}^{L} \sum_{k=1}^{K} \left| \mathbb{E}_{\mathcal{D}_\ell^k - \mathrm{P}^k}[\hat{\phi}_\ell^k - \phi_0^k] \right| \leq K t_2 \right) \geq 1 - \frac{1}{t_2^2} \sum_{\ell=1}^{L} \sum_{k=1}^{K} \frac{\|\hat{\phi}_\ell^k - \phi_0^k\|_{\mathrm{P}^k}^2}{|\mathcal{D}_\ell^k|}. \tag{D.10}$$

Choose $t_1 := \sqrt{\frac{2}{\epsilon} \sum_{k=1}^{K} \frac{\rho_{k,0}^2}{|\mathcal{D}^k|}}$ and $t_2 := \sqrt{\frac{2}{\epsilon} \sum_{\ell=1}^{L} \sum_{k=1}^{K} \frac{\|\hat{\phi}_\ell^k - \phi_0^k\|_{\mathbb{P}^k}^2}{|\mathcal{D}_\ell^k|}}$. Then, with a probability greater than $1 - \epsilon$,

$$\sum_{k=1}^{K} R_1^k \leq K \left( \sqrt{\frac{2}{\epsilon} \sum_{k=1}^{K} \frac{\rho_{k,0}^2}{|\mathcal{D}^k|}} + \sqrt{\frac{2}{\epsilon} \sum_{\ell=1}^{L} \sum_{k=1}^{K} \frac{\|\hat{\phi}_\ell^k - \phi_0^k\|_{\mathbb{P}^k}^2}{|\mathcal{D}_\ell^k|}} \right)$$

$$= K \sqrt{\frac{2}{\epsilon}} \left( \sqrt{\sum_{k=1}^{K} \frac{\rho_{k,0}^2}{|\mathcal{D}^k|}} + \sqrt{\sum_{\ell=1}^{L} \sum_{k=1}^{K} \frac{\|\hat{\phi}_\ell^k - \phi_0^k\|_{\mathbb{P}^k}^2}{|\mathcal{D}_\ell^k|}} \right).$$

### D.7.5 Proof of Theorem 4 - (3)

By Lemma D.3,

$$\mathbb{P}^k \left( \left| \mathbb{E}_{\mathcal{D}_\ell^k - \mathbb{P}^k} [\hat{\phi}_\ell^k - \phi_0^k] \right| > t \right) \leq \frac{1}{t^2} \frac{\|\hat{\phi}_\ell^k - \phi_0^k\|_{\mathbb{P}^k}^2}{|\mathcal{D}_\ell^k|}. \tag{D.11}$$

By Lemma D.4,

$$\mathbb{P}^k \left( \frac{1}{L} \sum_{\ell=1}^{L} \left| \mathbb{E}_{\mathcal{D}_\ell^k - \mathbb{P}^k} [\hat{\phi}_\ell^k - \phi_0^k] \right| \leq t \right) \geq 1 - \frac{1}{t^2} \sum_{\ell=1}^{L} \frac{\|\hat{\phi}_\ell^k - \phi_0^k\|_{\mathbb{P}^k}^2}{|\mathcal{D}_\ell^k|}. \tag{D.12}$$

Equivalently, by choosing $t = \sqrt{\frac{1}{\epsilon} \sum_{\ell=1}^{L} \frac{\|\hat{\phi}_\ell^k - \phi_0^k\|_{\mathbb{P}^k}^2}{|\mathcal{D}_\ell^k|}}$,

$$\frac{1}{L} \sum_{\ell=1}^{L} \left| \mathbb{E}_{\mathcal{D}_\ell^k - \mathbb{P}^k} [\hat{\phi}_\ell^k - \phi_0^k] \right| \overset{\text{w.p } 1-\epsilon}{\leq} \sqrt{\frac{1}{\epsilon} \sum_{\ell=1}^{L} \frac{\|\hat{\phi}_\ell^k - \phi_0^k\|_{\mathbb{P}^k}^2}{|\mathcal{D}_\ell^k|}}. \tag{D.13}$$

Define

$$A^k := \mathbb{E}_{\mathcal{D}^k - \mathbb{P}^k} [\phi_0^k] \tag{D.14}$$

$$B^k := \frac{1}{L} \sum_{\ell=1}^{L} \mathbb{E}_{\mathcal{D}_\ell^k - \mathbb{P}^k} [\hat{\phi}_\ell^k - \phi_0^k] \tag{D.15}$$

$$C^k := \frac{1}{L} \sum_{\ell=1}^{L} \left| \mathbb{E}_{\mathcal{D}_\ell^k - \mathbb{P}^k} [\hat{\phi}_\ell^k - \phi_0^k] \right| \tag{D.16}$$

$$\Delta_k := \sqrt{\frac{1}{\epsilon} \sum_{\ell=1}^{L} \frac{\|\hat{\phi}_\ell^k - \phi_0^k\|_{\mathbb{P}^k}^2}{|\mathcal{D}_\ell^k|}}. \tag{D.17}$$

Here,

$$R^k := A^k + B^k. \tag{D.18}$$

Then,

$$\mathbb{P}^k \left( R^k < x \right) \tag{D.19}$$

$$= \mathbb{P}^k \left( A_k + B_k < x \right) \tag{D.20}$$

$$= \mathbb{P}^k \left( A_k < x - B_k \right) \tag{D.21}$$

$$\leq \mathbb{P}^k \left( A_k < x + C_k \right) \tag{D.22}$$

$$\overset{\text{w.p } 1-\epsilon}{\leq} \mathbb{P}^k \left( A_k < x + \Delta_k \right). \tag{D.23}$$

Then,

$$\left| \mathrm{P}^k \left( A_k < x + \Delta_k \right) - \Phi(x) \right| \tag{D.24}$$

$$= \left| \mathrm{P}^k \left( A_k < x + \Delta_k \right) - \Phi(x + \Delta_k) + \Phi(x + \Delta_k) - \Phi(x) \right| \tag{D.25}$$

$$\leq \left| \mathrm{P}^k \left( A_k < x + \Delta_k \right) - \Phi(x + \Delta_k) \right| + \left| \Phi(x + \Delta_k) - \Phi(x) \right| \tag{D.26}$$

$$\leq \frac{0.4748\kappa_0^3}{\rho_{k,0}^3 \sqrt{|\mathcal{D}^k|}} + \left| \Phi(x + \Delta_k) - \Phi(x) \right| \qquad \text{(Prop. D.1)} \tag{D.27}$$

$$= \frac{0.4748\kappa_0^3}{\rho_{k,0}^3 \sqrt{|\mathcal{D}^k|}} + \left| \Phi'(x')\Delta_k \right| \qquad \text{(Mean-value theorem)} \tag{D.28}$$

$$\leq \frac{0.4748\kappa_0^3}{\rho_{k,0}^3 \sqrt{|\mathcal{D}^k|}} + \frac{1}{\sqrt{2\pi}}\Delta_k. \tag{D.29}$$

This completes the proof. ∎

### D.8  Proof for Corollary 4

By Cauchy-Schwartz' inequality,

$$\frac{1}{L}\sum_{\ell=1}^{L}\sum_{i=1}^{m}\mathbb{E}_{P^{i+1}}[\{\mu_0^i - \hat{\mu}_\ell^i\}\{\hat{\pi}_\ell^i - \pi_0^i\}] \leq \frac{1}{L}\sum_{\ell=1}^{L}\sum_{i=1}^{m}O_{P^{i+1}}\left(\|\mu_0^i - \hat{\mu}_\ell^i\|\|\pi_0^i - \hat{\pi}_\ell^i\|\right). \tag{D.30}$$

Given assumption, the upper bound in Eq. (15) converges at $o_{\mathrm{P}^k}(1/\sqrt{|\mathcal{D}_\ell^k|})$. Therefore, we conclude that $R^k$ converges in distribution to $\texttt{normal}(0, \rho_{k,0}^2)$.

## E  More Experiments

In this section, we demonstrate the DML-UCA estimator through examples for the ETT, $S$-admissibility, FD, Verma's equation, and Ctf-DE described in Sec. 2. For each example, the proposed estimator is constructed using a dataset $\mathrm{D}^k$ following a distribution $\mathrm{P}^k$. Our goal is to provide empirical evidence of the fast convergence behavior of the proposed estimator compared to competing baseline estimators. We consider two standard baselines in the literature: the 'regression-based estimator (reg)' only uses the regression nuisance parameters $\boldsymbol{\mu}$, and the 'ratio-based estimator (ratio)' that only uses the ratio nuisance parameters $\boldsymbol{\pi}$, while our DML-UCA estimator ('dml') uses both. Details of the regression-based ('reg') and the ratio-based ('ratio') estimators are provided in Sec. A. Details of experimental setting is provided in Sec. F. In this experiments, we set all variables other than the treatment variable $X$ as continuous.

We compare DML-UCA estimator to the regression-based estimator ('reg') and the ratio-based estimator ('ratio'). In particular, we use $\hat{\psi}^{\mathrm{est}}$ for est $\in \{\mathrm{reg}, \mathrm{pw}, \mathrm{dml}\}$ to denote the regression-based, probability-weighting, and DML-UCA estimators. We assess the quality of the estimators by computing the *average absolute error* $\mathrm{AAE}^{\mathrm{est}}$ which is defined as follow. For the ETT and Ctf-DE, $\mathrm{AAE}^{\mathrm{est}} := |\hat{\psi}^{\mathrm{est}} - \psi_0|$, where $\psi_0 := \mathbb{E}[Y_{X=0} \mid X = 1]$ for the ETT and $\psi_0 := \mathbb{E}[Y_{X=0,W_{X=1}} \mid X = 2]$ for the Ctf-DE. For the other examples, $\mathrm{AAE}^{\mathrm{est}} := \frac{1}{\mathrm{domqin}(X)}\sum_{x\in\mathrm{domain}(X)}|\hat{\psi}^{\mathrm{est}}(x) - \psi_0(x)|$ where $\psi_0(x) := \mathbb{E}[Y \mid \mathrm{do}(x)]$, $\hat{\psi}^{\mathrm{est}}(x)$ is an estimator for $\psi_0(x)$ and $\mathrm{dom}(X)$ is a cardinality of the domain of $X$. Nuisance functions are estimated using XGBoost (Chen and Guestrin, 2016). We ran 100 simulations for each number of samples $n = \{2500, 5000, 10000, 20000\}$ and drew the AAE plot. We evaluate the $\mathrm{AAE}^{\mathrm{est}}$ in the presence of the 'converging noise $\epsilon$' as in Sec. 4.

**Statistical Robustness.** The AAE plots for all scenarios are presented in Fig. E.4. For all examples, all the estimators ('reg', 'pw', 'dml') converge as the sample size grows. Furthermore, the proposed DML-UCA estimator outperforms the other two estimators by achieving fast convergence. This result

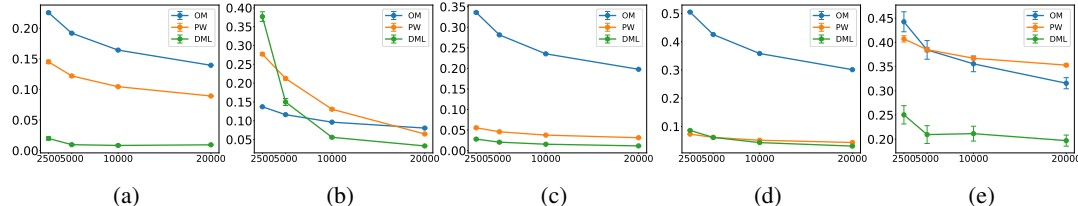

Figure E.4: **(a)** ETT in Sec. B, **(b)** Transportability ($S$-admissibility) in Sec. B, **(c)** Front-door in Example 1, **(d)** Verma in Example 2, **(e)** Ctf-DE in Example 3.

corroborates the robustness property in Thm. 4, which implies that DML-UCA converges faster than the other counterparts.

# F   Details in Experiments

As described in Sec. 4, we used the XGBoost (Chen and Guestrin, 2016) as a model for estimating nuisances. We implemented the model using Python. In modeling nuisance using the XGBoost, we used the command `xgboost.XGBClassifier(eval_metric='logloss')`[1] to use the XGBoost. We tuned the parameters for each examples to empirically guarantee the convergence of the regression and ratio nuisances. For each examples, the same parameters are used globally for implementing DML-UCA, regression-based estimator, ratio-based estimator, or other competing estimators (Fulcher et al., 2019; Jung et al., 2021a).

Now, we present the structural causal models (SCMs) utilized for generating the dataset. Furthermore, we include a segment of the code employed to generate the dataset.

## F.1   FD (Fig. 1a) for Simulation in Fig. 2a

We define the following structural causal models:

$$
\begin{aligned}
U &\sim \texttt{normal}(0.5, 0, 5),\\
U_{Z_i} &\sim \texttt{normal}(0, 1), \text{ for } i = 1, \cdots, d_Z\\
C_i &\coloneqq f_{C_i}(U), \text{ where } \mathbf{C} \coloneqq \{C_i : i = 1, \cdots, d_C\}\\
X &\coloneqq f_X(\mathbf{C}, U),\\
Z_i &\coloneqq f_{Z_i}(\mathbf{C}, X), \text{ where } \mathbf{Z} \coloneqq \{Z_i : i = 1, \cdots, d_Z\}\\
Y &\coloneqq f_Y(\mathbf{C}, Z, U),
\end{aligned}
$$

where

$$
\begin{aligned}
f_{C_i}(U) &\coloneqq \left\lfloor \frac{1}{1 + \exp(0.25 U_Z + 2U - 1)} \right\rfloor,\\
f_X(\mathbf{C}, U) &\coloneqq \texttt{Binary}\left( \frac{1}{1 + \exp(2\mathbf{C}^\intercal \mathbf{1} - 1 + U)} \right),\\
f_{Z_i}(\mathbf{C}, X) &\coloneqq \texttt{Binary}\left( \frac{1}{1 + \exp(2X - 1 + 0.5\mathbf{C}^\intercal \mathbf{1} + U_{Z_i})} \right)\\
f_Y(\mathbf{C}, Z, U) &\coloneqq \frac{1}{1 + \exp((1/d_C)\mathbf{C}^\intercal \mathbf{1} + (1/d_Z)(2\mathbf{Z}^\intercal \mathbf{1} - 1) + 2U)}.
\end{aligned}
$$

The parameterization for XGBoost used in $\boldsymbol{\mu}$ called (`mu_params`) and $\boldsymbol{\pi}$ called (`pi_params`) is the following:

---

[1]Detailed parametrization of parameters including learning rates, maximum depth of the trees, etc. are explained in `https://xgboost.readthedocs.io/en/stable/python/python_api.html#module-xgboost.training`.

```
   mu_params = {
 'booster': 'gbtree',
 'eta': 0.3,
 'gamma': 0,
 'max_depth': 10,
 'min_child_weight': 1,
 'subsample': 1.0,
 'colsample_bytree': 1,
 'lambda': 0.0,
 'alpha': 0.0,
 'objective': 'reg:squarederror',
 'eval_metric': 'rmse',
 'n_jobs': 4
 }

pi_params = {
  'booster': 'gbtree',
  'eta': 0.3,
  'gamma': 0,
  'max_depth': 10,
  'min_child_weight': 1,
  'subsample': 0.0,
  'colsample_bytree': 1,
  'objective': 'binary:logistic',
  'eval_metric': 'logloss',
  'reg_lambda': 0.0,
  'reg_alpha': 0.0,
  'nthread': 4
 }
```

### F.2  Verma (Fig. 1b) for Simulation in Fig. 2b

We define the following structural causal models:

$$U_{XB} \sim \texttt{normal}(1,0,5),$$
$$U_{AY} \sim \texttt{normal}(-1,0,5),$$
$$U_A \sim \texttt{normal}(0,1)$$
$$U_B \sim \texttt{normal}(0,1)$$
$$X := f_X(U_X B)$$
$$A_i := f_{A_i}(X, U_{AY}), \text{ for } i = 1, \cdots, d_A$$
$$B_i := f_{B_i}(X, U_{XB}), \text{ for } i = 1, \cdots, d_B$$
$$Y := f_Y(\mathbf{B}, U_{AY}),$$

where

$$f_X(U_X B) := \texttt{Binary}\left(\frac{1}{1 + \exp(2U_{XB} - 1)}\right),$$

$$f_{A_i}(X, U_{AY}) := \texttt{Binary}\left(\frac{1}{1 + \exp(2X - 1 + U_A + U_{AY})}\right)$$

$$f_{B_i}(X, U_{XB}) := \texttt{Binary}\left(\frac{1}{1 + \exp(2\mathbf{A}^\intercal\mathbf{1} - 1 + U_B + 0.5U_{XB})}\right)$$

$$f_Y(\mathbf{B}, U_{AY}) := \frac{1}{1 + \exp(2\mathbf{B}^\intercal\mathbf{1} - 1 + 0.5U_{AY})}.$$

The parameterization for XGBoost used in $\mu$ called (`mu_params`) and $\pi$ called (`pi_params`) is the following:

```
    mu_params = {
 'booster': 'gbtree',
 'eta': 0.35,
 'gamma': 0,
 'max_depth': 6,
 'min_child_weight': 1,
 'subsample': 1.0,
 'colsample_bytree': 1,
 'lambda': 0.0,
 'alpha': 0.0,
 'objective': 'reg:squarederror',
 'eval_metric': 'rmse',
 'n_jobs': 4  # Assuming you have 4 cores
}

pi_params = {
 'booster': 'gbtree',
 'eta': 0.1,
 'gamma': 0,
 'max_depth': 10,
 'min_child_weight': 1,
 'subsample': 0.0,
 'colsample_bytree': 1,
 'objective': 'binary:logistic',  # Change as per your objective
 'eval_metric': 'logloss',  # Change as per your needs
 'reg_lambda': 0.0,
 'reg_alpha': 0.0,
 'nthread': 4
}
```

## F.3   Example estimand (Fig. 1e) for Simulation in Fig. 2c

We define the following structural causal models:

$$U_{X_1,Z} \sim \texttt{normal}(1,0,5),$$
$$U_{X_1,Y} \sim \texttt{normal}(-1,0,5),$$
$$U_{Z,Y} \sim \texttt{normal}(0.5,0.5)$$
$$U_R \sim \texttt{normal}(0,0.5)$$
$$U_Z \sim \texttt{normal}(0,0.5)$$
$$U_{X_2} \sim \texttt{normal}(0,0.5)$$
$$X_1 := f_{X_1}(U_{X_1,Z}, U_{X_1,Y})$$
$$Z_i := f_{Z_i}(X_1, U_{X1,Z}, U_{Z,Y}), \text{ for } i = 1, \cdots, d_Z$$
$$R_i := f_{R_i}(X_1),, \text{ for } i = 1, \cdots, d_R$$
$$Y := f_Y(\mathbf{B}, U_{AY}),$$

where

$$f_{X_1}(U_{X_1,Z}, U_{X_1,Y}) := \texttt{Binary}\left(\frac{1}{1+\exp(2U_{X_1,Z}-U_{X_1,Y}-1)}\right),$$

$$f_{R_i}(X_1) := \texttt{Binary}\left(\frac{1}{1+\exp(2X_1-1+U_R)}\right)$$

$$f_{Z_i}(X_1, U_{X1,Z}, U_{Z,Y}) := \texttt{Binary}\left(\frac{1}{1+\exp(4X_1-1+U_Z+U_{X_1,Z}+U_{Z,Y})}\right)$$

$$f_{X_2}(\mathbf{Z}, X_1) := \texttt{Binary}\left(\frac{1}{1+\exp((2X_1-1)\mathbf{Z}^\intercal\mathbf{1}-U_{X_2})}\right),$$

$$f_Y(\mathbf{R}, X_2, U_{X_1,Y}, U_{Z,Y}) := \frac{1}{1+\exp((1/dR)\mathbf{R}^\intercal\mathbf{1}+2X_2-1+2(U_{X_1,Y}+U_{Z,Y}))}.$$

The parameterization for XGBoost used in $\mu$ called (`mu_params`) and $\pi$ called (`pi_params`) is the following:

```
    mu_params = {
 'booster': 'gbtree',
 'eta': 0.3,
 'gamma': 0,
 'max_depth': 8,
 'min_child_weight': 1,
 'subsample': 0.8,
 'colsample_bytree': 0.8,
 'lambda': 0.0,
 'alpha': 0.0,
 'objective': 'reg:squarederror',
 'eval_metric': 'rmse',
 'n_jobs': 4  # Assuming you have 4 cores
}

pi_params = {
 'booster': 'gbtree',
 'eta': 0.1,
 'gamma': 0,
 'max_depth': 10,
 'min_child_weight': 1,
 'subsample': 0.75,
 'colsample_bytree': 0.75,
 'objective': 'binary:logistic',  # Change as per your objective
 'eval_metric': 'logloss',  # Change as per your needs
 'reg_lambda': 0.0,
 'reg_alpha': 0.0,
 'nthread': 4
}
```

## F.4 ETT in Sec. B for Simulation in Fig. E.4a

We define the following structural causal models:

$$U_X \sim \mathtt{normal}(0,1)$$
$$U_Y \sim 0.5 \; texttt{normal}(0,1)$$
$$\mathbf{Z} \sim 0.25\mathtt{normal}(0,1,dZ),$$
$$X := f_X(\mathbf{Z})$$
$$Y := f_Y(X,\mathbf{Z})$$

where

$$f_X(\mathbf{Z}) := \mathtt{Binary}\left(\frac{1}{1+\exp(2\mathbf{Z}^\intercal\mathbf{1}-1+U_X)}\right)$$
$$f_Y(\mathbf{Z},X) := \frac{1}{1+\exp(\mathbf{Z}^\intercal\mathbf{1}(2X-1)+U_Y)}.$$

The parameterization for XGBoost used in $\mu$ called (`mu_params`) and $\pi$ called (`pi_params`) is the following:

```
    mu_params = {
 'booster': 'gbtree',
 'eta': 0.5,
 'gamma': 0,
 'max_depth': 15,
 'min_child_weight': 1,
 'subsample': 0.8,
 'colsample_bytree': 1,
 'lambda': 0,
 'alpha': 0,
 'objective': 'reg:squarederror',
 'eval_metric': 'rmse',
 'n_jobs': 4  # Assuming you have 4 cores
}

pi_params = {
 'booster': 'gbtree',
 'eta': 0.3,
 'gamma': 0,
 'max_depth': 10,
 'min_child_weight': 1,
 'subsample': 1,
 'colsample_bytree': 1,
 'objective': 'binary:logistic',  # Change as per your objective
 'eval_metric': 'logloss',  # Change as per your needs
 'reg_lambda': 1,
 'reg_alpha': 0,
 'nthread': 4
}
```

## F.5 Transportability in Sec. B for Simulation in Fig. E.4b

We define the following structural causal models:

$$U_X \sim \texttt{normal}(0, 1)$$
$$U_Y \sim 0.5\ texttt{normal}(0, 1)$$
$$\mathbf{Z} \sim 0.25\texttt{normal}(0, 0.5, dZ) + S\texttt{normal}(0.1, 0.5, dZ)$$
$$X := f_X(\mathbf{Z})$$
$$Y := f_Y(X, \mathbf{Z})$$

where

$$f_X(\mathbf{Z}) := \texttt{Binary}\left(\frac{1}{1 + \exp((1/dZ)(2\mathbf{Z}^\intercal\mathbf{1} - 1) + U_X)}\right)$$

$$f_Y(\mathbf{Z}, X) := \frac{1}{1 + \exp(\mathbf{Z}^\intercal\mathbf{1}(2X - 1) + U_Y)}.$$

The parameterization for XGBoost used in $\mu$ called (`mu_params`) and $\pi$ called (`pi_params`) is the following:

```
    mu_params = {
 'booster': 'gbtree',
 'eta': 0.3,
 'gamma': 0,
 'max_depth': 15,
 'min_child_weight': 1,
 'subsample': 0.8,
 'colsample_bytree': 1,
 'lambda': 0,
 'alpha': 0,
 'objective': 'reg:squarederror',
 'eval_metric': 'rmse',
 'n_jobs': 4  # Assuming you have 4 cores
}

pi_params = {
 'booster': 'gbtree',
 'eta': 0.1,
 'gamma': 0,
 'max_depth': 10,
 'min_child_weight': 1,
 'subsample': 1,
 'colsample_bytree': 1,
 'objective': 'binary:logistic',  # Change as per your objective
 'eval_metric': 'logloss',  # Change as per your needs
 'reg_lambda': 1,
 'reg_alpha': 0,
 'nthread': 4
}
```

## F.6 FD with continuous mediators for Simulation in Fig. E.4c

We define the following structural causal models:

$$\mathbf{U_C} \sim \mathtt{normal}(0, 1, d_C)$$
$$U \sim \mathtt{normal}(0, 1)$$
$$\mathbf{C} := f_{\mathbf{C}}(U)$$
$$X := f_X(U, \mathbf{C})$$
$$Z := f_Z(X, \mathbf{C})$$
$$Y := f_Y(U, Z, \mathbf{C})$$

where

$$f_{\mathbf{C}}(U) := 0.25\mathbf{U_C} + 2U - 1$$
$$f_X(U, \mathbf{C}) := \mathtt{Binary}\left(\frac{1}{1 + \exp((2\mathbf{C^\intercal 1} - 1) + U)}\right)$$
$$f_Z(X, \mathbf{C}) := \frac{1}{1 + \exp(0.1\mathbf{C^\intercal 1}(2X - 1) + X)}$$
$$f_Y(\mathbf{Z}, X) := \frac{1}{1 + \exp(\mathbf{C^\intercal 1} + (2Z - 1) + U)}.$$

The parameterization for XGBoost used in $\mu$ called (`mu_params`) and $\pi$ called (`pi_params`) is the following:

```
    mu_params = {
 'booster': 'gbtree',
 'eta': 0.01,
 'gamma': 0,
 'max_depth': 10,
 'min_child_weight': 1,
 'subsample': 1.0,
 'colsample_bytree': 1,
 'lambda': 0.0,
 'alpha': 0.0,
 'objective': 'reg:squarederror',
 'eval_metric': 'rmse',
 'n_jobs': 4
}

pi_params = {
 'booster': 'gbtree',
 'eta': 0.3,
 'gamma': 0,
 'max_depth': 20,
 'min_child_weight': 1,
 'subsample': 0.0,
 'colsample_bytree': 1,
 'objective': 'binary:logistic',
 'eval_metric': 'logloss',
 'reg_lambda': 0.0,
 'reg_alpha': 0.0,
 'nthread': 4
}
```

### F.7 Verma's equation with continuous mediators for Simulation in Fig. E.4d

We define the following structural causal models:

$$U_{XB} \sim \texttt{normal}(1, 0, 5),$$
$$U_{AY} \sim \texttt{normal}(-1, 0, 5),$$
$$X := f_X(U_X B)$$
$$A := f_A(X, U_{AY})$$
$$B := f_B(X, U_{XB})$$
$$Y := f_Y(B, U_{AY}),$$

where

$$f_X(U_X B) := \texttt{Binary}\left(\frac{1}{1 + \exp(2U_{XB} - 1)}\right),$$

$$f_A(X, U_{AY}) := \texttt{Binary}\left(\frac{1}{1 + \exp(2X - 1 + 0.5U_A Y)}\right)$$

$$f_B(X, U_{XB}) := \texttt{Binary}\left(\frac{1}{1 + \exp(2A - 1 + 0.5U_X B)}\right)$$

$$f_Y(B, U_{AY}) := \frac{1}{1 + \exp(2B - 1 + 0.5U_A Y)}.$$

The parameterization for XGBoost used in $\mu$ called (`mu_params`) and $\pi$ called (`pi_params`) is the following:

```
mu_params = {
 'booster': 'gbtree',
 'eta': 0.35,
 'gamma': 0,
 'max_depth': 6,
 'min_child_weight': 1,
 'subsample': 1.0,
 'colsample_bytree': 1,
 'lambda': 0.0,
 'alpha': 0.0,
 'objective': 'reg:squarederror',
 'eval_metric': 'rmse',
 'n_jobs': 4  # Assuming you have 4 cores
}

pi_params = {
 'booster': 'gbtree',
 'eta': 0.1,
 'gamma': 0,
 'max_depth': 10,
 'min_child_weight': 1,
 'subsample': 0.0,
 'colsample_bytree': 1,
 'objective': 'binary:logistic',  # Change as per your objective
 'eval_metric': 'logloss',  # Change as per your needs
 'reg_lambda': 0.0,
 'reg_alpha': 0.0,
 'nthread': 4}
```

## F.8 Ctf-DE in Example 3 for Simulation in Fig. E.4e

We define the following structural causal models:

$$U \sim \texttt{normal}(0, 2),$$
$$X := f_X(U)$$
$$Z := f_Z(U)$$
$$W := f_W(X, Z)$$
$$Y := f_Y(X, Z, W),$$

where

$$f_X(U) := \begin{cases} 0 \text{ if } \frac{1}{1+\exp(2U_{XB}-1)} < 0.5 \\ 1 \text{ if } \leq 0.5 \frac{1}{1+\exp(2U_{XB}-1)} < 0.8 \\ 2 \text{ if } \leq 0.8 \frac{1}{1+\exp(2U_{XB}-1)}. \end{cases}$$

$$f_Z(U) := \frac{1}{1 + \exp(-U + 1)}$$

$$f_W(X, Z) := \frac{1}{1 + \exp(X - 1 + Z)}$$

$$f_Y(Z, X, W) := \frac{1}{1 + \exp(3X - 1 + 0.1Z + 0.1W + W(X - 1))}.$$

The parameterization for XGBoost used in $\mu$ called (`mu_params`) and $\pi$ called (`pi_params`) is the following:

```
mu_params = {
 'booster': 'gbtree',
 'eta': 0.3, # vab
 'gamma': 0.0,
 'max_depth': 6, #vb (same as va)
 'min_child_weight': 1,
 'subsample': 1.0,
 'colsample_bytree': 1,
 'lambda': 0.0,
 'alpha': 0.0,
 'objective': 'reg:squarederror',
 'eval_metric': 'rmse',
 'n_jobs': 4  # Assuming you have 4 cores
}

pi_params = {
 'booster': 'gbtree',
 'eta': 0.05,
 'gamma': 0,
 'max_depth': 10,
 'min_child_weight': 1,
 'subsample': 1.0,
 'colsample_bytree': 1,
 'objective': 'multi:softprob',  # Change as per your objective
 'num_class': 3,
 'eval_metric': 'mlogloss',  # Change as per your needs
 'reg_lambda': 0.0,
 'reg_alpha': 0.0,
 'nthread': 4
}
```

