# OpenReview forum: "Unified Covariate Adjustment for Causal Inference"
_NeurIPS.cc/2024/Conference — NeurIPS 2024 poster_

### Official Review · Reviewer_cBfm · 2024-06-24

**Soundness:** 4
**Presentation:** 3
**Contribution:** 4
**Rating:** 7
**Confidence:** 3

**Summary:**

The paper introduces a new framework for identifying causal estimands, referred to as Unified Covariate Adjustment (UCA).

It demonstrates that the UCA-expressible class (a class of causal estimands identifiable by UCA) is extensive, encompassing estimands identified by the (sequential) back-door adjustment, front-door adjustment, and Tian’s adjustment.

Furthermore, the paper proposes an estimation strategy for the UCA-expressible class using machine learning methods and analyzes the error of such estimators.

Notably, the proposed estimator is scalable and achieves the double robustness property.

**Strengths:**

1. The paper provides good examples to illustrate the relationship between the UCA-expressible class and other classes of estimands identified by different adjustments.

2. It offers comprehensive identification and estimation strategies, thereby presenting a complete causal inference methodology.

3. The theoretical analysis of the estimator's scalability and error characterization is solid.

**Weaknesses:**

1. While the paper provides sufficient conditions for an estimand being not UCA-expressible (i.e., necessary conditions for an estimand being UCA-expressible), it lacks necessary conditions for an estimand being not UCA-expressible (i.e., sufficient conditions for an estimand being UCA-expressible). However, addressing this issue might be beyond the scope of this paper.

2. In the presence of unmeasured confounders, causal estimands (e.g., average treatment effects) may not be identifiable by UCA since identification requires bridge functions and a specifically designed ID algorithm (e.g., Shpitser et al. 2023, JMLR, The Proximal ID Algorithm). It may be worth mentioning these points at the end of Section 2, along with the "napkin" estimand.

3. The authors claimed the estimator is doubly robust (page 8, line 307) under certain conditions, which includes $n^{-1/4}$ convergence rate for $\widehat{\mu}^i$. However, I believe this condition may not be achievable, as the error tends to accumulate for large $i$; see Question below.

**Questions:**

page 2, Table 1: Can the authors clarify or provide examples of why UCA does not cover the functionals identified by obsID/gID?

page 3, line 94: There are double commas.

page 4, line 145: Given that the probability is at most 1, does this mean that $r_i$ is discrete? If so, can the authors explain why $r_i$ has to be discrete?

Section 3.1:

- Estimation of $\mu$:
I think the error $\| \widehat{\mu}^i - \mu_0^i \|$ is not only affected by the estimation errors from the $i$th stage, but also by the estimation errors from the previous stages. In other words, the estimation error of $\mu$ accumulates as the index $i$ increases. Can the authors provide some comments on this error accumulation?

- Bias structure:
The bias structure $R_2$ in Theorem 3 consists of [error of $\mu^i$]$\times$[error of $\pi^i$] and [error of $\mu^i$]$\times$[error of $\pi^{i-1}$]. The second cross-$i$ product term seems non-trivial. Can the authors provide an example of causal estimands that only has the bias structure of [error of $\mu^i$]$\times$[error of $\pi^i$] without cross-$i$ product terms?

Sections F.1.4 and F.1.5: Typo: textttnormal

**Limitations:**

Limitations are not explicitly discussed in the paper.

The authors might consider including the weaknesses mentioned above as limitations.

---

> ### Author Rebuttal · Authors · 2024-08-06
>
> We thank the reviewer for the time and valuable feedback, and appreciate the positive assessment of our work.
>
> ---
>
> > While the paper provides sufficient conditions for an estimand being not UCA-expressible (i.e., necessary conditions for an estimand being UCA-expressible), it lacks necessary conditions for an estimand being not UCA-expressible (i.e., sufficient conditions for an estimand being UCA-expressible). However, addressing this issue might be beyond the scope of this paper.
>
> As mentioned, developing necessary conditions is challenging and beyond the scope of this paper. We believe this question has the potential to open a new direction for enhancing the proposed method. Thank you for the insightful question.
>
> ---
>
> > In the presence of unmeasured confounders, causal estimands (e.g., average treatment effects) may not be identifiable by UCA since identification requires bridge functions and a specifically designed ID algorithm (e.g., Shpitser et al. 2023, JMLR, The Proximal ID Algorithm). It may be worth mentioning these points at the end of Section 2, along with the "napkin" estimand.
>
> In Table 1, we provide the coverage of the UCA, which states that not all obsID/gID functions are identified. We will further mention this point as suggested at the end of Section 2.
>
>
> ---
>
> > The authors claimed the estimator is doubly robust (page 8, line 307) under certain conditions, which includes $n^{-1/4}$  convergence rate for $\hat{\mu}^i$. However, I believe this condition may not be achievable, as the error tends to accumulate for large $i$; see Question below.
> > Estimation of $\mu$: I think the error $\|\hat{\mu}^i - \mu^i_0 \|$  is not only affected by the estimation errors from the $i$th stage, but also by the estimation errors from the previous stages. In other words, the estimation error of $\mu$ accumulates as the index $i$ increases. Can the authors provide some comments on this error accumulation?
>
> Good point. Due to the nature of nested expectation (and nested regression), the error of the nuisances can accumulate as $i$ decreases from $m$ to $1$. The term $\|\hat{\mu}^i - \mu^i_0 \|$ indeed represents the accumulated error of $\hat{\mu}^i$. Even with this accumulation, the error decomposes into the product of the errors of the nuisances; i.e., $\text{[error of DML]} = \sum_{i=1}^{m} \text{[error of $\mu^i$]} \times \text{[error of $\pi^i$]}$ still holds. As long as $\hat{\mu}^i$ converges to $\mu^i_0$ (which is likely in practice with flexible ML models), even if errors accumulate, the rate of convergence of the DML estimator outperforms that of competing estimators (OM, PW).
>
> On the other hand, we note that $n^{-1/4}$ is used to exemplify the debiasedness property because it’s the fastest rate that a neural network can achieve [1] .
>
> [1] Györfi, László, et al. “A distribution-free theory of nonparametric regression” (2002)
>
> ---
>
> > page 2, Table 1: Can the authors clarify or provide examples of why UCA does not cover the functionals identified by obsID/gID?
>
> UCA does not cover all the functionals identified by obsID/gID. We provide an example called Napkin estimand in lines 202-204, with a causal diagram in Figure 1c. The identification estimand is given as $\frac{ \sum_{w}P(x,y \mid r,w)P(w) }{ \sum_{w}P(x \mid r,w)P(w) }$. The UCA cannot handle cases where the functional is given as a ratio of two functions. A detailed discussion on the coverage of the UCA is provided in Section C.3.
>
>
> > page 4, line 145: Given that the probability is at most 1, does this mean that $r_i$ is discrete? If so, can the authors explain why $r_i$  has to be discrete?
>
> No, it means that the value of $R_i$ is governed by the probabilistic measure $\sigma^i$. Whether $R_i$ is continuous or discrete depends on the choice of $\sigma^i$. For example, if $\sigma^i$ is a uniform distribution over $[a,b]$, then $R_i$ can be any real number within $[a,b]$. However, if $\sigma^i$ is a Bernoulli distribution, then $R_i$ is a $\{0,1\}$ binary variable.
>
> ---
>
> > Bias structure: The bias structure $R_2$  in Theorem 3 consists of [error of $\mu^i$] $\times$ [error of $\pi^{i}$] and [error of $\mu^i$] $\times$ [error of $\pi^{i-1}$]. The second cross-$i$ product term seems non-trivial. Can the authors provide an example of causal estimands that only has the bias structure of [error of $\mu^i$] $\times$ [error of $\pi^i$] without cross-$i$ product terms?
>
> Consider the back-door adjustment where $i=1$. Since the error term only contains [error of $\mu\^i$] $\times$ [error of $\pi\^i$], the second cross-term doesn’t exist when $i=1$.
>
> ---
>
> > Sections F.1.4 and F.1.5: Typo: textttnormal
> > page 3, line 94: There are double commas.
>
> Thank you for catching the typo. These will be fixed.
>
> ---
>
> > The authors might consider including the weaknesses mentioned above as limitations.
>
> We will discuss more about the limitations based on provided feedback. Thanks.

---

> > ### Comment · Reviewer_cBfm · 2024-08-10
> >
> > Thank you for your response. I believe the paper is well-prepared for acceptance, so I will keep my score as is.

---

> > > ### Author Response · Authors · 2024-08-11
> > > **Response**
> > >
> > > Thank you for taking the time and effort to provide constructive feedback.

---

### Official Review · Reviewer_ZBfF · 2024-07-03

**Soundness:** 4
**Presentation:** 3
**Contribution:** 3
**Rating:** 7
**Confidence:** 3

**Summary:**

This paper describes the estimand framework "unified covariate adjustment (UCA)" and discusses its coverage with multiple examples (Front-door, Verma's equation, Counterfactual directed effect and most importantly Tian's adjustment). Then it develops an estimator for this function class and shows that it is scalable. Lastly the authors run experiments on simulated data to empirically demonstrate the robustness and the scalability of their estimator.

**Strengths:**

* This work is original as it introduces - to the best of my knowledge - the only scalable estimator for many functions classes. It makes a lot of novel causal inference results applicable in real applications.

* The overall quality of this work is very good, the results are provided as theoretically sound theorems and empirically confirmed through simulated data experiments.

* This paper is clear, the contributions are clearly stated and the paper is well structured with some useful examples.

**Weaknesses:**

* All the proofs are provided in the supplementary material and the intuition for the proofs are not provided in the main paper.

* Some further experiments could be interesting even they are not necessary. For example: how does DML compare to prior scalable estimators on BD/SBD? In figure 2a,b,c how much can the dimenion of the summed variables grow before the running time of DML reaches unreasonable values (eg. 2000)? Similar question for figure2d,f how little can thee sample size be before the errors of DML reaches unreasonable values?

* UCA could be more thoroughly delimited. It is well defined and the authors give examples of scenarios that are included, how they do not clearly discuss which type Ctf-ID are covered and which are not (This is also true for obsID/gID and transportability). Furthermore, there is no discussions concerning non-covered function classes and what difficulties prevent estimation in those cases.

* Some intuition regarding the meaning of the mathematical objects would have been appreciated (eg. what are the sets $\bm{C}_i$ and $\bm{R}_i$ in Def 1)?

* While pseudo-code is provided, giving access to the code is always appreciated.

**Questions:**

* Could you give intuition concerning the meaning of the $\bm{C}_i$ and $\bm{R}_i$ and $\bm{S}_i$ in Def 1 as well as $\bm{\mu}$ and $\bm{\pi}$ for non experts.

* Could you discuss the limits of DML, ie. when the dimension grows and when the sample size shrinks.

* Concerning the experiments, each point corresponds to the average of 100 simulations. Could you provide the variances as well?

**Limitations:**

* The authors do not discuss which function classe are not covered by UCA. Moreover, they do not discuss the limits of their estimator DML when the dimension grows and when the sample size shrinks.

---

> ### Author Rebuttal · Authors · 2024-08-06
>
> We thank the reviewer for the time and valuable feedback, and appreciate the positive assessment of our work.
>
> ---
>
> > Some further experiments could be interesting even they are not necessary. For example: how does DML compare to prior scalable estimators on BD/SBD? In figure 2a,b,c how much can the dimenion of the summed variables grow before the running time of DML reaches unreasonable values (eg. 2000)? Similar question for figure2d,f how little can thee sample size be before the errors of DML reaches unreasonable values?
>
> We have provided a set of experimental results in the attached pdf file.
>
> ---
>
> > UCA could be more thoroughly delimited. It is well defined and the authors give examples of scenarios that are included, how they do not clearly discuss which type Ctf-ID are covered and which are not (This is also true for obsID/gID and transportability). Furthermore, there is no discussions concerning non-covered function classes and what difficulties prevent estimation in those cases.
>
> Some cases where the target estimand cannot be expressed through UCA are discussed in Appendix C.3. A summary of this discussion will be provided in the main paper.
>
> ---
>
> > Some intuition regarding the meaning of the mathematical objects would have been appreciated (eg. what are the sets $C_i$ and $R_i$ in Def 1)?
> > Could you give intuition concerning the meaning of the $C_i$ and $R_i$ and $S_i$  in Def 1 as well as $\mu$ and $\pi$ for non experts.
>
> The meaning of $C_i$, $R_i$, and $S_i$ depends on specific cases. In all examples, we specified what $C_i$, $R_i$, and $S_i$ meant. For the BD/SBD, we can view $C_i$ as a set of covariates, $R_i$ as a treatment, and $S_i$ as predecessors of $C_{i+1}$. $\mu$ is a (nested-) expectation functional representing the UCA estimand, and $\pi$ is the probability-weighting-based functional representing the UCA estimand. We will provide more explanation to give an intuition about these mathematical objects in the paper. Thank you.
>
> ---
>
> > While pseudo-code is provided, giving access to the code is always appreciated.
>
> We will make the code available after the revision. Thank you.
>
> ---
>
> > Could you discuss the limits of DML, ie. when the dimension grows and when the sample size shrinks.
>
> As shown in the experiment in the PDF of the global response, the proposed DML estimator remains scalable when the dimension is high.
>
> When the sample size shrinks, it’s possible that the error of the DML estimator is amplified because its error decomposes into the product of the errors of nuisances; i.e., $\text{[error of DML]} = \sum_{i=1}^{m} \text{[error of } \mu^i \text{]} \times \text{[error of } \pi^i \text{]}$. If the sample is small so that the errors of nuisances become large, the resulting DML estimator may have a larger error since the error is multiplied. However, as the sample size grows, the DML estimator is guaranteed to converge faster whenever nuisances are converging to the truth.
>
> ---
>
> > Concerning the experiments, each point corresponds to the average of 100 simulations. Could you provide the variances as well?
>
> In all plots in Figure 2, the confidence intervals of the error with $\alpha = 0.05$ are shown as error bars.
>
> ---
>
> > The authors do not discuss which function classe are not covered by UCA.
>
> Some function classes where the target estimand cannot be expressed through UCA are discussed in Appendix C.3.
>
> We will add a sufficient criterion to determine which estimands can be represented as a UCA estimand in the revision of the paper. The idea behind the criterion is as follows: If
>
> 1. The target estimand is expressed as the mean of the product of conditional distributions over $(\mathbf{C}_1, \mathbf{R}_1, \cdots, \mathbf{C}_m, \mathbf{R}_m)$; and
>
> 2. The variables that are marginalized and fixed simultaneously (e.g., $X$ in the front-door adjustment) only appear in $\mathbf{C}_1$,
>
> the proposed methods can be applied. These conditions are sufficient for applying the empirical bifurcation technique (Def 2) that allows scalable estimation.

---

> > ### Comment · Reviewer_ZBfF · 2024-08-08
> > **Response to author's rebuttal**
> >
> > Thank you for all the clarifications. After reading other reviews, I still think it is an interesting paper and I maintain my score.

---

> > > ### Author Response · Authors · 2024-08-11
> > > **Response**
> > >
> > > Thank you for spending the time reading our paper and the positive assessment.

---

### Official Review · Reviewer_Z7av · 2024-07-12

**Soundness:** 2
**Presentation:** 1
**Contribution:** 3
**Rating:** 6
**Confidence:** 3

**Summary:**

The paper presents a class of adjustment formulas called unified covariate adjustment (UCA) which is shown to be able to express many classes of adjustments known in the existing literature. A scalable and doubly robust estimator for UCA is also presented along with some experimental results.

**Strengths:**

The proposed UCA estimator seems to be very expressive and is able to model many existing estimators in the literature. Examples were given to show how existing estimators can be expressed as a UCA estimator.

The paper also proposes a doubly robust method to obtain UCA estimates.

**Weaknesses:**

- It is of my *personal opinion* that the paper lacks polish. There are many instances of technical notations being used without first properly defining them, making it hard to understand and follow the discussion (see the Questions section for some of them). The reader should not be expected to guess the meaning of certain notations by cross-referencing across subsequent pages (at best check the preliminaries/notation section) or even across other paper references.
- The paper claims on Lines 57-58 that "while these estimators are designed to achieve a wide coverage of functionals, they lack scalability due to the necessity of summing over high-dimensional variables" but the general definition of UCA in equation (1) also involves summing over potentially many variable values in $c \cup r$. Please explain clearly why UCA avoids scalability issues.
- Line 302-304: It is not true that only asymptotic analyses were known for all these estimators. For example, [1] gives non-asymptotic finite sample guarantees for the Tian-Pearl adjustment. The paper would benefit from a comparison against such prior works and illustrate how DML-UCA adjustments are indeed more sample efficient as compared to existing estimators.
- Why is there no experiments against the Tian-Pearl adjustment?
- No code was released (though some parameters were given the appendix).

[1] Arnab Bhattacharyya, Sutanu Gayen, Saravanan Kandasamy, Vedant Raval, Vinodchandran N. Variyam Proceedings of The 25th International Conference on Artificial Intelligence and Statistics, PMLR 151:7531-7549, 2022.

**Questions:**

- How are the symbols in Table 1 determined? "scalable" is defined to be "evaluable in polynomial time relative to number of covariates and capable in the presence of mixed discrete and continuous covariates", but about "coverage"?
- Consider rephrasing the awkward-sounding sentence "Our work strives maximizing coverage..." on Line 73-76?
- $S^b_{i-1}$ first appears in Line 141. How is this defined? How does it differ from $S_{i-1}$?
- In Line 144, $S_0, C_0, R_0$ are referenced but how are they defined? Also, how do the $S_i$s relate to the variable set $V$? Is $S_i$ the c-component of the $i^{th}$ vertex?
- Equation (4): The notation $v^{(i−1)}$ is undefined. Is $v^{(i−1)} = \\{v_1, \ldots, v_i\\}$ as in the Tian-Pearl 2002 paper?
- Line 312: What is $O_{P^{i+1}}$? How does it differ from just the usual big-O notation? Also, how do the errors in the estimation scale with the number of samples? Lines 307-309 only say "**if** the terms converge at a rate of $n^{-1/4}$, then DML-UCA coverges at a rate of $n^{-1/2}$". Why do those terms converge at a rate of $n^{-1/4}$?
- Given a general causal graph and causal query, how does one find a suitable and valid UCA expression? What is the procedure?

Potential typos:
- Double comma on Line 94
- Extra ) on Line 153

**Limitations:**

Nil

---

> ### Author Rebuttal · Authors · 2024-08-06
>
> Thank you for your feedback and for the opportunity to provide further elaboration.
>
> ---
>
> > technical notations being used without first properly defining them
>
> We will further proofread the paper and the preliminaries.
>
> ---
>
> > Please explain clearly why UCA avoids scalability issues.
>
> Existing BD/SBD estimators avoid scalability issues by replacing marginalization with nested expectation. However, when a variable is both marginalized and fixed simultaneously (e.g., FD: $\sum_{x’,c}E[ Y | z,x’,c] P(z | x, c) P(x’,c)$, where $X$ is fixed to $x$ in $P(z | x,c)$ and marginalized by $\sum_{x’}$ in other components), representing the marginalization operator as a nested expectation is non-trivial. These challenges lead to potential scalability issues for previous FD estimators (Fulcher et al., 2019; Guo et al., 2023), which have a complexity of $O(n2^m + T(n,m))$ ($m$: the dimension of variables, $T(n,m)$: time complexity of learning nuisances). In contrast, the newly proposed UCA estimator uses empirical bifurcation to replace marginalization with nested expectation, achieving $O(n + T(n,m))$ by leveraging the empirical bifurcation method.
>
> ---
>
> > [1] gives non-asymptotic finite sample guarantees for the Tian-Pearl adjustment.
>
> We will cite all the referred papers. Please note that lines 302-304 discuss DML-style estimators, while the mentioned paper provides finite sample guarantees for a basic plug-in estimator under the discrete random variable setting.
>
> ---
>
> > Why is there no experiments against the Tian-Pearl adjustment?
>
> Figure 2(b,e) shows the experimental results for Verma's graph (Figure 1b), which is an example instance of the Tian-Pearl adjustment.
>
> ---
>
> > No code was released (though some parameters were given the appendix).
>
> We will make the code available after the revision. Thank you.
>
> ---
>
> > How are the symbols in Table 1 determined? "scalable" is defined to be "evaluable in polynomial time relative to number of covariates and capable in the presence of mixed discrete and continuous covariates", but about "coverage"?
>
> Coverage indicates whether established estimators exist. For obsID/gID classes, estimators like those by Jung et al. (2021a, 2023a), Xia et al. (2021, 2022), and Bhattacharya et al. (2022) are marked in the "Prior" column. The UCA, covering Tian's adjustment, is checked in its respective column.
>
> ---
>
> > Consider rephrasing the awkward-sounding sentence "Our work strives maximizing coverage..." on Line 73-76?
>
> Thank you. We will rephrase as follows: “Our work aims to maximize coverage, enabling the effective development of scalable estimators with the doubly robust property.”
>
> ---
>
> > $S^b_{i-1}$ first appears in Line 141. How is this defined? How does it differ from $S_{i-1}$?
>
> $\mathbf{S}^b_{i-1}$ represents a set of variables fixed to $\mathbf{s}^b_{i-1}$ in a conditional distribution $P^i(\mathbf{V}) = Q^i(\mathbf{V} \mid \mathbf{S}^b_{i-1} = \mathbf{s}^b_{i-1})$ (e.g., $\mathbf{S}^b_1 = \{X\}$ with $\mathbf{s}^b_1 = \{x\}$ in FD, Example 1). Meanwhile, $\mathbf{S}_{i-1}$ is a subset of the union of $\mathbf{C}^{(i-1)}$ and $\mathbf{R}^{(i-1)}$, excluding the fixed set.
>
> ---
>
> > $S_0, C_0, R_0$  are referenced but how are they defined? Also, how do the $S_i$s relate to the variable set $V$? Is $S_i$  the c-component of the $i$th  vertex?
>
> 1. $S_0$, $C_0$, and $R_0$ are all defined as the empty set. We will add this in the preliminaries.
> 2. We define $\mathbf{S}\_{i-1} := (\mathbf{C}\^{(i-1)} \cup \mathbf{R\}^{(i-1)} ) \setminus \mathbf{S}\^{b}\_{i-1}$ in line 144. This is a subset of the variable set $\mathbf{V}$.
> 3. Thank you for the good question. $\mathbf{S}_i$ is not a c-component. We only used $\mathbf{S}\_{X}$ to denote the c-component containing $\{X\}$. We will improve the notation to distinguish them more explicitly.
>
> ---
>
> > Equation (4): The notation $v^{(i-1)}$ is undefined. Is $v^{(i-1)} = \{v_1,\cdots,v_i\}$  as in the Tian-Pearl 2002 paper?
>
> Yes, it's defined in line 98.
>
> ---
>
> > Line 312: What is $O_{P^{i+1}}$?  How does it differ from just the usual big-O notation? Also, how do the errors in the estimation scale with the number of samples?
>
> > Why do those terms converge at a rate of $n^{-1/4}$?
>
> 1. As written in line 104, $O_P$ is a stochastic boundedness called the big-O in probability [van der Vaart, "Asymptotic Statistics."] (1998). The expression $f(\mathbf{V}) = O_{P^{i+1}}(n^{-1/4})$ means that $n^{1/4} \times f(\mathbf{V})$ will be bounded even when $n$ increases to infinity. This indicates that $f(\mathbf{V})$ decreases at least as fast as $n^{1/4}$. If the error term is $O_{P^{i+1}}(n^{-1/4})$, then it decreases at the rate of $n^{1/4}$.
> 2. Theorem 3 and Corollary 3 show that when nuisances converge at the rate of $n^{-\alpha}$ ($\forall \alpha \in (0,1)$), the estimator can converge at the double rate: $n^{-2\alpha}$. We demonstrate this with $\alpha = 1/4$, since $n^{-1/4}$ is the fastest convergence rate for modern ML models like neural networks [1].
>
> [1] Györfi, László, et al. “A distribution-free theory of nonparametric regression” (2002)
>
> ---
>
> > how does one find a suitable and valid UCA expression?
>
> Some causal queries that satisfy known graphical criteria (e.g. BD/SBD, FD, or Tian’s adjustment) can be represented as an UCA. On the other hand, we have further developed the sufficient criterion to determine if a given estimand can be represented as a valid UCA expression. The idea behind the criterion is as follows: If
>
> 1. The target estimand is expressed as the mean of the product of conditional distributions over $(\mathbf{C}\_1, \mathbf{R}\_1, \cdots, \mathbf{C}\_m, \mathbf{R}\_m)$; and
>
> 2. The variables that are marginalized and fixed simultaneously (e.g., $X$ in the front-door adjustment) only appear in $\mathbf{C}_1$,
>
> the proposed methods can be applied. This criterion will be added in the revised version of the paper.
>
> ---
>
> > Double comma on Line 94, Extra ) on Line 153
>
> We will fix the typos, thank you.

---

> > ### Comment · Reviewer_Z7av · 2024-08-10
> >
> > Thank you for the detailed responses. They have addressed my concerns. I look forward to these being incorporated nicely in a future revision. I will increase my score upwards.

---

> > > ### Author Response · Authors · 2024-08-11
> > > **Response**
> > >
> > > We appreciate your constructive input. Thank you for the positive feedback.

---

### Official Review · Reviewer_Q5Gy · 2024-07-13

**Soundness:** 3
**Presentation:** 3
**Contribution:** 2
**Rating:** 7
**Confidence:** 3

**Summary:**

The paper introduces a novel framework, unified covariate adjustment (UCA), which covers a broad class of sum-product causal estimands and additionally develops a scalable estimator (via DML-UCA) that ensures double robustness.

**Strengths:**

* The paper presents a well-developed theoretical framework with clear assumptions and derivations. The motivation to extend existing estimands is well-articulated, and the comparisons with prior work are thoroughly examined.

* The paper is well-written and clearly presents the motivation, methodology, and contributions.

* It is interesting to revisit the coverage and scalability of previous studies and provide comprehensive evaluations.

**Weaknesses:**

* The proposed method is based on structural causal models, which have been extensively studied.  UCA-class is an extension of the sequential back-door adjustment (SBD), and there are already existing studies that address similar questions

* The authors mention in Example 1 that the Front-Door adjustment (FD) can be represented using the UCA framework. Are there any assumptions required to validate this representation? Similarly, to represent the Verma constraints as UCA in Example 2, are there any criteria that could be followed to verify the representation in practice? What are the requirements and limitations for implementing these representations to ensure the reliability and validity of the estimation in general?

**Questions:**

See Weaknesses

**Limitations:**

yes

---

> ### Author Rebuttal · Authors · 2024-08-05
>
> Thank you for sharing your thoughts and feedback!
>
> > UCA-class is an extension of the sequential back-door adjustment (SBD), and there are already existing studies that address similar questions
>
> Indeed, UCA is an extension of the SBD, and we have appreciated and cited papers regarding estimating the SBD estimand. However, naively applying the SBD estimators to the UCA-class (e.g., FD, Verma) may lead to biased estimators whenever the value of SBD estimand and the UCA estimand don't match. Also, modifying the existing SBD estimators to the UCA-class is non-trivial since variables that are fixed and marginalized at the same time (e.g., ‘X’ in FD) are not properly treated in the SBD estimators. A special method (such as the empirical bifurcation in Def. 6) is required to develop an estimator. Furthermore, for Tian’s adjustment, the weighting nuisances $\pi^{i}_0$ do not have the same form as those in SBD. In summary, developing doubly robust estimators for the UCA class is a novel and non-trivial task.
>
> ---
>
> > Are there any assumptions required to validate this representation? Similarly, to represent the Verma constraints as UCA in Example 2, are there any criteria that could be followed to verify the representation in practice? What are the requirements and limitations for implementing these representations to ensure the reliability and validity of the estimation in general?
>
> Some causal queries that satisfy known graphical criteria (e.g. BD/SBD, FD, or Tian’s adjustment) can be represented as an UCA. Recall that representing the Tian's adjustment through the UCA estimand is demonstrated in the paper.
>
> On the other hand, we have further developed the sufficient criterion to determine if a given estimand can be represented as a valid UCA expression. The idea behind the criterion is as follows: If
>
> 1. The target estimand is expressed as the mean of the product of conditional distributions over $(\mathbf{C}_1, \mathbf{R}_1, \cdots, \mathbf{C}_m, \mathbf{R}_m)$; and
>
> 2. The variables that are marginalized and fixed simultaneously (e.g., $X$ in the front-door adjustment) only appear in $\mathbf{C}_1$ (that is,  $\mathbf{S}^{b}\_{i-1} \cap \mathbf{C}\^{\geq 2} = \emptyset$),
>
> then the proposed methods can be applied. These conditions are sufficient for applying the empirical bifurcation technique (Def 2) that allows scalable estimation.

---

> > ### Comment · Reviewer_Q5Gy · 2024-08-12
> >
> > Thanks for the detailed responses, I will maintain my score.

---

> > > ### Author Response · Authors · 2024-08-12
> > > **Response**
> > >
> > > Thank you for your positive assessment of our paper.

---

### Author Rebuttal · Authors · 2024-08-06

We attached a PDF to report the experimental results in respond to the following questions from Reviewer ZBfF:

1. > In figure 2a,b,c how much can the dimenion of the summed variables grow before the running time of DML reaches unreasonable values (eg. 2000)?

2. > how little can thee sample size be before the errors of DML reaches unreasonable values?

In summary, the proposed DML-UCA estimator can be evaluated under a high-dimensional setting where $d = 50000$. The estimator can also be evaluated under a small sample size setting, with samples varying from 10 to 100.

---

### Decision · Program_Chairs · 2024-09-25

**Decision:**

Accept (poster)

**Comment:**

The paper introduces a new causal identification framework which covers a wide range of estimands. The authors provided detailed (and ultimately satisfactory) responses to the initial reviews and all reviewers have recommended acceptance.